# TOY MODELS OF COMBINATORIAL INTERPRETABILITY

We introduce *combinatorial interpretability*, a methodology for understanding neural computation by analyzing the combinatorial structures in the sign-based categorization of a network's weights and biases. We demonstrate its power through *feature channel coding*, a theory that explains how neural networks compute Boolean expressions and potentially underlies other categories of neural network computation. According to this theory, features are computed via feature channels: unique cross-neuron encodings shared among the inputs the feature operates on. Because different feature channels share neurons, the neurons are polysemantic and the channels interfere with one another, making the computation appear inscrutable. We show how to decipher these computations by analyzing a network's feature channel coding, offering complete mechanistic interpretations of several small neural networks that were trained with gradient descent. Crucially, this is achieved via static combinatorial analysis of the weight matrices, without examining activations or training new autoencoding networks. It also allows us for the first time to exactly quantify and explain the relationship between a network's parameter size and its computational capacity (the set of features it can compute with low error), a relationship that is implicitly at the core of many modern scaling laws.

## 1 INTRODUCTION

Recent research in neural network interpretability often views computation through high-dimensional activation spaces, analyzing directions corresponding to computed features. This geometric perspective has revealed phenomena such as polysemanticity, the superposition hypothesis Elhage et al. (2022), and strong empirical evidence of correlations between features and activation vector directions Ameisen et al. (2025b;a); Bricken et al. (2023); Cunningham et al. (2024); Dunefsky et al. (2024b;a); Ge et al. (2024); Makelov et al. (2024); Marks et al. (2025); Rajamanoharan et al. (2024a;b). While these techniques ingeniously capture feature representations and some interdependencies, they have yet to fully elucidate a complete dependency graph, the underlying functions being computed by the networks, or their precise computational mechanisms.

This paper introduces a combinatorial approach to interpretability, contrasting with the prevailing geometric view of activation spaces. Our combinatorial view focuses on the sign based categorization (positive, negative, or zero) of learned parameters, rather than their precise values. [1] While not mutually exclusive with the vector space perspective, this combinatorial approach offers insights into facets of interpretability—such as the logic of the circuits that compute features—that are challenging for vector space methods. We demonstrate, initially on small examples but with potential for future scalability, that our approach can directly extract features and their exact computations from the network's learned parameter structure. This extraction occurs without requiring auxiliary models like sparse autoencoders or transcoders Ameisen et al. (2025a); Dunefsky et al. (2024b); Ge et al. (2024); Marks et al. (2025); Rajamanoharan et al. (2024a). Essentially, our work aims to reveal the low-level computational mechanics—the how—going beyond merely identifying the when and where of feature relationships.

We focus on the computation of Boolean expressions—a meaningful class that facilitates initial exploration of such combinatorial representations. This allows us to propose a theory explaining how neural networks, at least for Boolean formulae, compute by representing features and their computations as combinatorial structures

---

[1]One could categorize to more than three classes, but we are starting with the simplest, which as you will see, already exposes lots of structure.

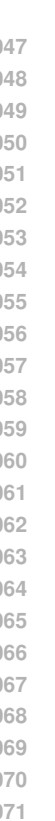
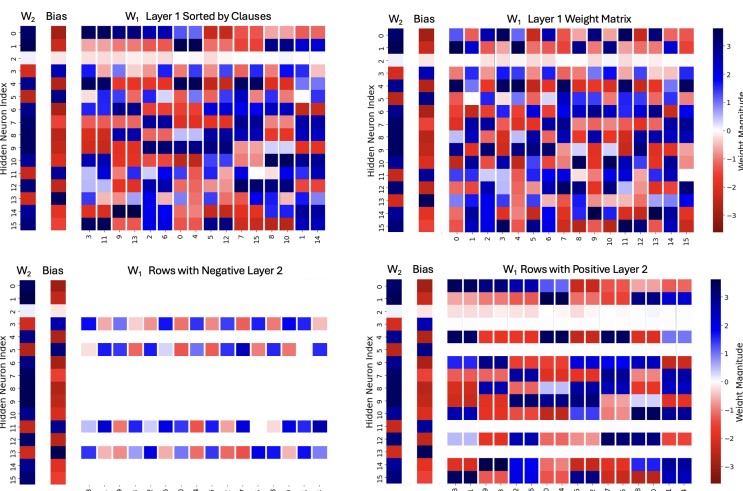

Figure 1: All weights and biases for a neural network trained on the Boolean formula
$(x_3 \wedge x_{11}) \vee (x_9 \wedge x_{13}) \vee (x_2 \wedge x_6) \vee (x_0 \wedge x_4) \vee (x_5 \wedge x_{12}) \vee (x_7 \wedge x_{15}) \vee (x_8 \wedge x_{10}) \vee (x_1 \wedge x_{14})$

encoded in their weights and biases. Specifically, our theory explains phenomena like polysemanticity and superposition as arising from feature-associated codes.

Central to our combinatorial approach is the *feature channel coding hypothesis*. This hypothesis specifies how features are represented (analogous to the superposition hypothesis) and, crucially, how computation proceeds between them (a novel aspect of our theory). Under this hypothesis, a feature in a given layer is represented by a "code"—a specific subset of neurons at that layer. The "bits" of this code identify the neurons whose activation contribute to that feature being active. Minimal overlap between the codes of different features at the same layer results in nearly orthogonal activation vectors (when considering precise values), mirroring the superposition hypothesis.

The key differentiator of feature channel coding is its explanation of how computation works. To compute a feature $f_i$, all prior-layer input features necessary for its computation are mapped to $f_i$'s code. The network then performs the computation for $f_i$ on each neuron in this code. For example, Boolean $f_i = x_{i_1} \wedge x_{i_2}$ is computed as $\text{ReLU}(x_{i_1} + x_{i_2} - 1))$. Thus, when the required input features from the prior layer are active, each neuron in $f_i$'s code activates via its individual computation, collectively activating feature $f_i$. If a prior-layer feature contributes to multiple current-layer features, it is mapped to each of their respective codes. Crucially, a feature's code is primarily a mechanism for routing and congregating its necessary inputs, rather than directly encoding the function itself. While code overlap can introduce noise, computations remain robust if this overlap is minimal, yielding a sufficiently good approximation to the target result. Identifying these codes provides a description of the network's computation directly from its weight matrices, bypassing the need to learn neuron directionality in activation space.

Before continuing, we wish to acknowledge that our work, like Anthropic's Elhage et al. (2022)), uses greatly simplified settings without showing how they extend to general LLMs. The problem of fully interpreting neural computation is hard, which is why we are initially focusing on much simpler settings in which we can show full interpretation for the first time. The hope is that these can be used to establish principles that are applicable in the large.

## 2   NEURAL NETWORKS LEARN FEATURE CHANNEL CODING

An open question from Adler & Shavit (2024) is whether feature channel codes occur naturally in networks trained via gradient descent; in other words, do SGD-trained neural networks learn to utilize such coding? We demonstrate that these codes indeed emerge naturally from gradient descent training. To illustrate this we first consider a two-layer MLP configured with 16 inputs, 16 first-layer neurons (each with a bias), and a single second-layer neuron (without bias). The network was trained to learn various hidden Boolean functions. Initially we studied randomly selected functions from the set of all possible pairwise ANDs of 16 Boolean variables, where variables were paired such that each appears in exactly one AND. We here present a typical result, where the network was trained[2] on the formula described in Figure 1.

Figure 1 (upper right) displays the complete set of learned parameters from a representative training run: Layer 1 weights ($W_1$), Layer 1 bias, and Layer 2 weights ($W_2$, oriented upright for readability). Colors indicate weight sign (blue: positive, red: negative). Let us first examine $W_1$. In its raw, unaltered form (columns corresponding to input features $x_i$, rows to first-layer neurons), $W_1$ offers little immediate insight, beyond neuron 2 appearing largely inactive. However, Figure 1 (upper left) shows $W_1$ with its columns sorted by features, grouping variables that are conjoined by an AND (e.g., columns 3 and 11 for $(x_3 \wedge x_{11})$). This sorted view begins to reveal an interesting pattern: in many, though not all, positions, adjacent columns corresponding to these AND-pairs have similar values.

In Figure 1 (lower right), $W_1$'s features are sorted identically, but we now filter to include only rows (neurons) corresponding to positive Layer 2 weights (this filtering is justified below). This visualization is striking: in every row, the columns for each AND pair are virtually identical. Furthermore, observing the positive (blue) weights reveals that every pair of input variables appearing in the same AND clause is represented by a feature channel code. The features computed by the first layer of this network are exactly the pairwise ANDs of the Boolean formula. The neurons comprising the codes for each of those features allows the AND features' computation to propagate to the next layer where the OR of those features is computed. For example, input features $x_3$ and $x_{11}$, used to compute $(x_3 \wedge x_{11})$, are each mapped to the feature channel code consisting of positive weights on neurons $\{0, 4, 6, 10, 12, 15\}$. Similarly, the feature $(x_9 \wedge x_{13})$ is represented by the code $\{0, 6, 8, 14\}$, and $(x_5 \wedge x_{12})$ by $\{6, 8, 9, 10, 15\}$. We give a formal definition of feature channel coding in the appendix. Essentially, different features $f_i, f_j$ may have overlapping codes $C(f_i) \cap C(f_j) \neq \emptyset$, leading to polysemantic neurons. The collection $\{C(f) : f \in \mathcal{F}\}$ constitutes the *feature channel coding* of the layer, representing how the network routes subsets of input features onto shared groups of neurons for computation.

Figure 1 reveals that every neuron $k$ computes $x_i \wedge x_j$ with soft Boolean logic via $a_k() = \text{ReLU}(x_i + x_j - b)$ for some $b$ (regular Boolean variables would have $b = 1$). For every row (neuron) with a positive Layer 2 weight, the Layer 1 bias is significantly negative, representing this '$-b$' threshold. Thus, if two input features in the same AND clause (e.g., $(x_3 \wedge x_{11})$) are both True, their sum (calculated by the dot product of neuron weights with the input) will overcome the negative bias. This yields a positive value after the ReLU for all neurons in the pair's feature channel code, consequently leading to a "True" output from the network's sigmoid.

One advantage of the combinatorial approach is the ability to detect patterns that would be hard to define and track for vectors in high dimensional geometric space. In Figure 5 we used this property to track how feature channel codes emerge during training with SGD. We trained for 20 epochs a network with 32 neurons on a DNF with 4 clauses with 4 variables per clause, and looked at the emergence of patterns in the weights associated with the clauses as the network learns. The left side is the initial 0-epoch heat map of $W_1$, the middle is the final $W_1$ after 20 epochs, with the $x$-axes sorted by the variables of the 4-clauses. We look at

---

[2]Specifically, the network was trained using 30,000 random inputs, each equally probable as a True or False instance of the formula. Each input contained between two and six True variables, also chosen with equal probability. The training utilized the BCE loss function and the Adam optimizer with a learning rate of 0.01, conducted for 2000 epochs.

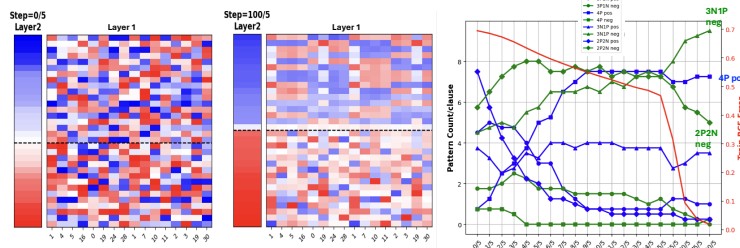

Figure 2: Emergence of codes and decrease in error during 20 epochs of training. Initial state left, final state middle, and the change in average code pattern per clause from initial to final on the right.

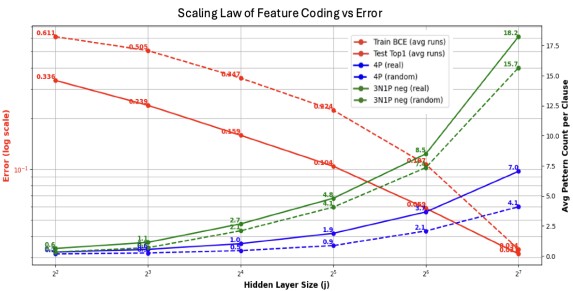

Figure 3: Feature channel coding provides combinatorial explanation for scaling as hidden layer size increases.

the 4-weight patterns for each clause and categorize them into five types based on the number of positive (P) and negative (N) weights corresponding to a clause's four variables in a given row (neuron). We explain this in detail in the appendix, but here note the clear patetrn: the number of positive 4P patterns corresponding to a soft AND of 4 variables rapidly increased, while other positive patterns diminished, and this in complete correlation with the error in training.

## 3   FEATURE CHANNEL CODING CAN EXPLAIN SCALING LAWS

In Figure 7, we demonstrate how the approach enables quantification of features, allowing us to use feature channel coding to explain traditional scaling laws: we keep the number of features/clauses fixed at $k = 64$ and increase the size of the hidden layer along the $x$-axis. We plot both the error and the frequency of 4P codes in positive rows (blue) and 3N1P codes in negative rows (green) versus the size of the hidden layer. We see that 4P and 3N1P code frequency (solid lines) is much higher than random (dotted lines) and that the decrease in error is tightly correlated to this increase in code frequency. We believe that this relationship is due to the capacity of the network to perform feature channel coding, where, as before, the network roughly achieves that limit.[3]

Our methods, which we will discuss in the workshop, extend beyond structure, enabling recovery of the underlying Boolean formula directly from learned parameters. For lack of space, we delegate to the appendix empirical stuidies of how feature channel coding explains computation in "superposition" Adler & Shavit (2024) and other results.

---

[3]Again, with a sufficiently large hidden layer, the network does not need to take full advantage of the space.

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

## A  NEURAL NETWORKS LEARN FEATURE CHANNEL CODING

The feature channel coding mechanism described earlier motivates the classification of $W_1$ rows based on the sign of their corresponding $W_2$ weights (visualized in Figure 1, bottom panels). The sign of a $W_2$ weight dictates whether its associated Layer 1 neuron's activation contributes positively or negatively to the network's final output. We call Layer 1 neuron activations that are positively weighted by $W_2$ *positive witnesses*; they provide evidence supporting a True instance with a strong positive signal. Conversely, we call activations negatively weighted by $W_2$ *negative witnesses*, providing evidence for a False instance through a negative signal. The visualization (Figure 1, lower left) suggests that negative witnesses effectively compute an XOR of variables within a target clause: if one variable of the pair is present but its counterpart is absent, it signals a False instance.

Interestingly, our methods extend beyond structure, enabling recovery of the underlying Boolean formula directly from learned parameters. In the pairwise AND case, the Layer 1 weight matrix suffices: take absolute values, then for each column $i$ find the column $j$ with highest correlation; if $j$ is top-correlated with $i$, we infer the pair $(i, j)$. (Results need not be symmetric.) Figure 4 illustrates performance when varying hidden size, formula size, and training set size. For each configuration we trained ten random formulas, measured test error on new inputs, and compared against the correlation algorithm's pairing accuracy (error bars show standard deviation). With sufficiently large training data, the algorithm almost always reconstructs the formula, and "sufficiently large" coincides with when the network itself achieves low test error. Appendix H.2 further demonstrates this approach for the four-consecutive-ones problem.

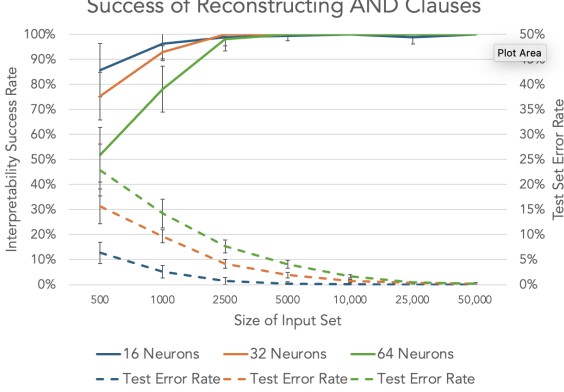

Figure 4: Formula recovery from Layer 1 weights

## B  FEATURE CHANNEL CODES EMERGE DURING TRAINING

One advantage of the combinatorial approach is the ability to detect patterns that would be hard to define and track for vectors in high dimensional geometric space. We used this advantage to see if and how feature channel coding emerges (Figure 5) during training with SGD. We study the alignment between Boolean formula clauses and specific patterns of positive and negative values within the Layer 1 weight matrix.

For clarity we show here one specific example execution though we conducted many that conform with what is seen here. We trained for 20 epochs a network with 32 neurons on a DNF with 4 clauses with 4 variables per clause, and looked at the emergence of patterns in the weights associated with the clauses as the network learns. The left side is the initial 0-epoch heat map of $W_1$, the middle is the final $W_1$ after 20 epochs, with

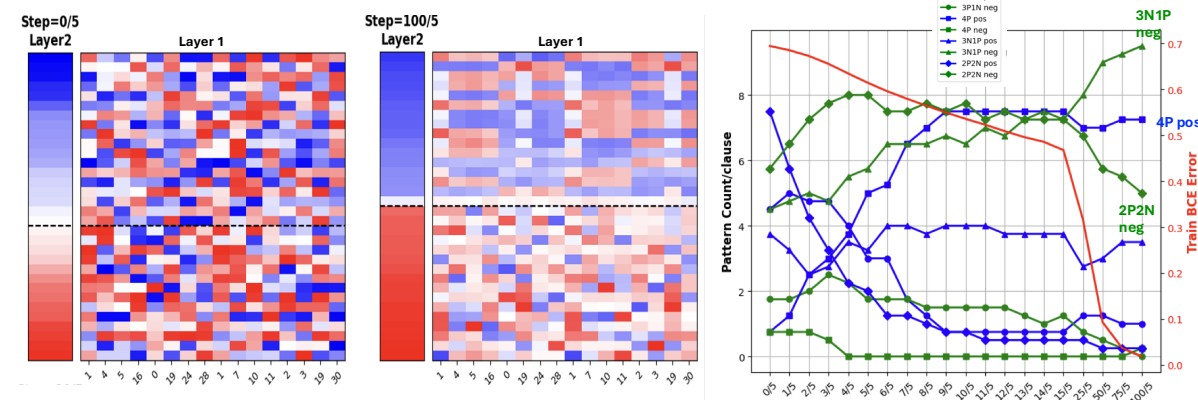

Figure 5: Emergence of codes and decrease in error during 20 epochs of training. Initial state left, final state middle, and the change in average code pattern per clause from initial to final on the right.

the $x$-axes sorted by the variables of the 4-clauses. We look at the 4-weight patterns for each clause and categorize them into five types based on the number of positive (P) and negative (N) weights corresponding to a clause's four variables in a given row (neuron): 4P (4 positive), 4N (4 negative), 3P1N (3 positive, 1 negative), 3N1P (3 negative, 1 positive), and 2P2N (2 positive, 2 negative). For example, 3P1N signifies that a clause's four variables align with a weight matrix row containing three positive and one negative value. The graph on the right depicts the various pattern counts per clause, the error in red. The $x$-axis non-linearly shows the first 3 epochs in more detail in 1/5 epoch intervals. Initally, in the random initial state, 4P and 4N patterns are the least prevalent, as expected about 1 per clause, and 2P2N is the most prevalent, close to 8 per clause. During training, positive 4P patterns rapidly increased, while other positive patterns diminished. Similarly, negative patterns, except 3P1N and 2P2N, decreased. By epoch 2, the dominant 4P positive coding pattern that we already showed corresponds an AND stabilized. However, training (BCE) error remained around 0.55 until negative witness 2P2N patterns transitioned to 3N1P, which was critical for achieving final accuracy. The final pattern (middle heatmap where clear 4P positive and 3N1P negative patterns are visible) is similar to the results of the trained graph of Figure 6 which we describe shortly. Notably, the rapid rise of 4P positive patterns before the first epoch underscores their importance and ease of discovery by gradient descent. While positive patterns converged quickly, the subsequent refinement of negative row patterns (flipping 2P2N to 3N1P) was essential for solving the task, demonstrating a multi-stage learning process where different pattern types become critical at different phases of training. "Why and how gradient descent converges to feature channel codes" is a fascinating research topic that the combinatorial setting could enable.

Additional results are in the appendix. In particular Appendix E presents a formal model of combinatorial interpretability and initial steps for disentangling multi-layer superposed neural networks. Appendix F examines single layer networks with multiple outputs. Appendix G provides more details on the quantitative analysis from Section C. Appendix H, offers more examples of feature channel coding and combinatorial network interpretation. Finally, we outline future research directions.

## C   COMBINATORIAL ANALYSIS OF A SCALING LAW

This section employs our combinatorial interpretability framework to investigate scaling laws governing *how* models compute Boolean formulas and *why* they fail with increasing complexity.

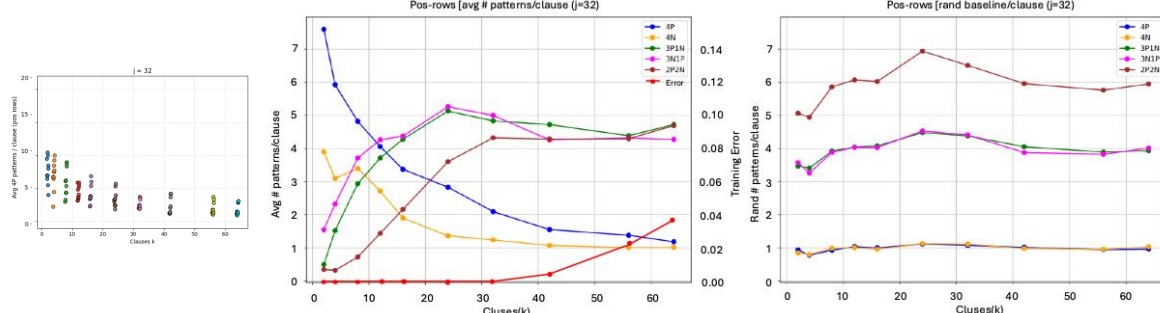

Figure 6: Prevalence of coding patterns in positive rows. The 4P pattern, in a blue line in middle plot, emerges as totally non-random. For better clarity on error, instead of error bars, the actual distribution of the ten 4P counts are given on the left. As can be seen, when there is sufficient room for multiple coding rows per clause, there is some variability on the size of the codes, but as code capacity becomes a constraint, the distribution becomes more concentrated.

We focus on DNF formulas composed of 4-variable clauses where all variables are positive (results for clauses with 3 positive and 1 negated variable are in Appendix G.4). As depicted in Figure 6, we achieve this by progressively increasing the number of clauses in the formula, starting from a small number $k = 2$, until the model can no longer learn it accurately with $k = 64$ clauses). This modification increases the complexity of the underlying truth table of the Boolean formula, leaving the network architecture and input size unchanged. Specifically, all experiments use a network with the structure described above with 32 input variables, 32 neurons, and a single sigmoid output neuron. All results are averaged over 10 trials, each with a distinct random Boolean formula. Further training details and deeper insights into network learning are provided in Appendix G.

Figure 6 provides quantitative evidence for feature channel coding and illuminates the network's learning capabilities and limitations. As before, each pattern type is represented by a line in the figure, where the vertical axis indicates the average number of positive witness Layer 1 weight matrix rows (associated with positive Layer 2 weights) exhibiting that specific pattern for a given clause. The middle graph in the figure shows results for positive witness rows, indicating that with few clauses, 4P patterns are abundant (e.g., over 7 per clause for $k = 2$), but their frequency declines as $k$ increases, with error (red) rising correspondingly. Error growth begins once the average 4P count per clause drops below $\sim$1.5–2, consistent with the feature coding hypothesis: below this threshold, clauses cannot be reliably coded. This decline stems from the weight matrix's limited capacity to both (a) allocate many 4P patterns across clauses and (b) preserve $\sim$50% negative weights per row. Increasing the fraction of positive weights would retain 4P patterns but compromise discrimination between four positives within one clause versus spread across clauses.

The middle graph of Figure 6 compares trained networks to those with random Layer 1 weights. Trained pattern frequencies clearly deviate from chance: patterns enriched relative to random aid learning, while depleted ones are uninformative. At low $k$, both 4P and 4N patterns are far more frequent than random; as $k$ grows, 4N frequency converges to random around $k = 30$, while 4P frequency, though reduced from $> 7\times$ random, remains above random even at $k = 64$. This supports the feature coding hypothesis: 4P patterns encode clauses across multiple rows, while 4N patterns offset them, enabling discrimination between four positives within one clause versus across clauses. Additional evidence, including prevalence of 3N1P codes in negative witness rows, is provided in the appendix.

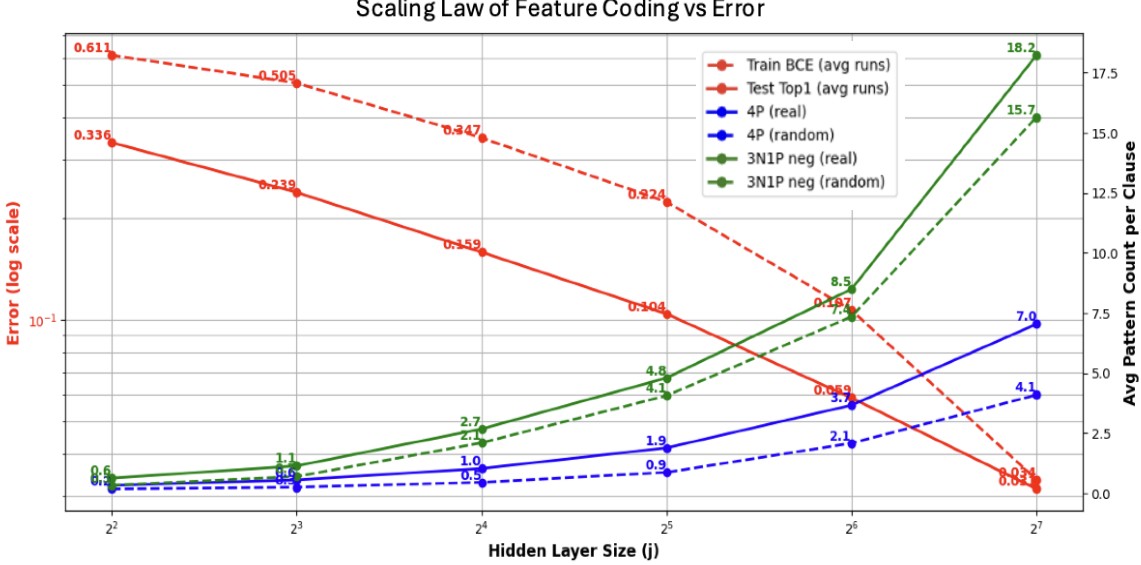

Figure 7: Feature channel coding provides combinatorial explanation for scaling as hidden layer size increases.

Next, in Figure 7, we demonstrate a similar capacity limit from a more traditional scaling law viewpoint: we keep the number of features/clauses fixed at $k = 64$ and increase the size of the hidden layer along the $x$-axis. We plot both the error and the frequency of 4P codes in positive rows (blue) and 3N1P codes in negative rows (green) versus the size of the hidden layer. We see that 4P and 3N1P code frequency (solid lines) is much higher than random (dotted lines) and that the decrease in error is tightly correlated to this increase in code frequency. We believe that this relationship is due to the capacity of the network to perform feature channel coding, where, as before, the network roughly achieves that limit.[4] This provides an easily explainable scaling law: the network's size implies a capacity to code the feature channels, with smaller networks being more limited in their coding ability, which explains the higher error rate.[5]

## D    BACKGROUND AND RELATED WORK

Early research in neural network interpretability viewed neurons as geometric vectors in activation space, associating neurons with single semantic directions Nguyen et al. (2016); Simonyan et al. (2013). However, it became evident that neurons often encode multiple unrelated features, known as *polysemanticity* Cunningham et al. (2023); Elhage et al. (2022;?); Goh et al. (2021); Scherlis et al. (2022), complicating interpretation Goodfellow et al. (2016); Haklay et al. (2025). This phenomenon relates closely to *superposition* Elhage et al. (2022), where there are more features than neurons. Polysemantic neurons have been empirically observed in large architectures, such as multimodal models like CLIP, where neurons simultaneously respond to disparate concepts Goh et al. (2021). This overlap represents an intentional strategy for efficient capacity allocation Elhage et al. (2022). This capacity allocation problem was isolated and studied in Scherlis et al. (2022). Theoretical analyses of combinatorial coding highlight this efficiency through explicit constructions Adler & Shavit (2024); Hänni et al. (2024); Mendel & Bushnaq (2024) and lower bounds Adler & Shavit (2024).

---

[4]Again, with a sufficiently large hidden layer, the network does not need to take full advantage of the space.

[5]Note that we trained here to half the epochs of prior examples and there is a larger error per model size.

Recent advances in the use of autoencoders for network interpretability have led to enhanced methods such as sparse autoencoders (SAEs) Cunningham et al. (2023) and transcoders Dunefsky et al. (2024b) for isolating feature vectors in activation space Ameisen et al. (2025b;a); Bricken et al. (2023); Cunningham et al. (2024); Dunefsky et al. (2024a); Ge et al. (2024); Makelov et al. (2024); Marks et al. (2025); Rajamanoharan et al. (2024a;b). These techniques allow the creation of *attribution graphs* that map the dependencies among features for the computation of a specific input Ameisen et al. (2025a). Yet, these attribution graphs are best described as targeted approximations of the original model, rather than elucidating the underlying computational processes. They capture which features influence subsequent features *for a specific input*, but they do not capture (a) the overall feature dependence (across all inputs), (b) the function that is being computed to realize that dependence, and (c) the underlying mechanism for computing that function.

Furthermore, these SAE-like techniques require new trained components that inherently simplify or omit certain aspects of the complexity of the original model. While this helps in understanding the feature landscape, it is fundamentally a reconfiguration rather than a direct observation of the original model's internal representation. In addition, recent work has raised doubts concerning the generalization ability of such approaches Heindrich et al. (2025). Consequently, while SAE-like techniques are valuable for gaining insights into model behavior, they do not capture all the details and interactions of the original model. In contrast, our combinatorial interpretabiliy approach, though at a much earlier phase, aims to directly interpret the computation from analyzing the structure of the model itself.

Previous interpretability work has reversed engineered the underlying computational mechanics of specific functions, although they do not reveal complete circuits as we do in this work. Research on 2SAT Palumbo et al. (2024) showed that transformers solving these Boolean formulae learn internal circuits that resemble symbolic algorithms. Nanda et al. Nanda et al. (2023) observed a discrete Fourier transform-like algorithm in networks performing modular arithmetic. Furthermore, theoretical frameworks such as the Information Bottleneck Tishby & Zaslavsky (2015) and statistical mechanics approaches Carleo et al. (2019) offer complementary insights into how neural representations encode data and manage correlations. Superposition is also connected to scaling laws, suggesting that larger models might reduce polysemanticity by developing more specialized neuron representations as resources increase Henighan et al. (2023); Kaplan et al. (2020); Rosenfeld et al. (2020).

## E    MODELING COMPUTATION AND COMBINATORIAL DISENTANGLEMENT

Thus far we have illustrated the combinatorial view of a two-layer network. Real-world models—even plain multilayer perceptrons (MLPs)—are usually richer: they begin with an embedding layer and then stack several nonlinear layers. This section introduces a conceptual model for understanding MLP computation based on underlying "feature circuits." We then propose a novel perspective viewing MLP layers as implicitly performing a decode-compute-encode cycle, aiming to disentangle the network and reveal this circuit. Building on this, we outline a potential strategy, termed *cascading feature disentanglement*, for systematically uncovering the feature circuit layer by layer. We present initial steps demonstrating how the first layer of computation can be revealed via disentanglement from an input embedding and also propose a general approach for deeper layers.

### E.1    THE FEATURE CIRCUIT VIEW OF NEURAL COMPUTATION

Our modeling follows the intuitive ideas in Adler & Shavit (2024); Elhage et al. (2022); Vaintrob et al. (2024). We aim to capture the fundamental computational structure implemented by an MLP. We model the computation at each layer $i$ (for $i = 1, \ldots, d$) as producing a set of abstract output features $F_i = \{f_{i,1}, f_{i,2}, \ldots, f_{i,m_i}\}$. Each feature $f_{i,j} \in F_i$ is computed by applying a specific function to a subset of the features $F_{i-1} = \{f_{i-1,1}, \ldots, f_{i-1,m_{i-1}}\}$ from the preceding layer (where $F_0$ represents the initial input

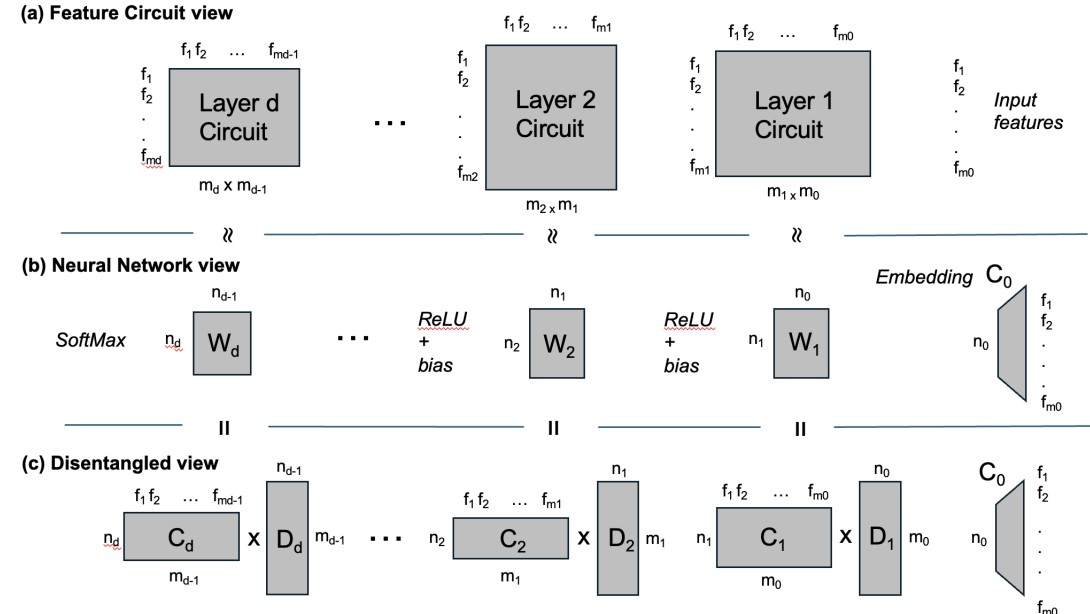

Figure 8: The feature circuit coding-decoding view of computation. Computation flows from right to left. The middle sequence (b) is the actual computation through the neural network. The top sequence (a) is the abstract computation mapping features to features in each layer being represented by the neural network. The lower sequence (c) is the feature coding view that bridges the two by thinking of the the weight matrix as implementing a decoding of the prior layer's features to serve as monosemantic inputs to a coding-and-computation of the features of the new one. $W_i = C_i D_i$ is a superposed $n_i \times n_{i-1}$ representation of the $m_i \times m_{i-1}$ $i$-th feature circuit layer.

features, $m_0$ in number). The complete *feature circuit* of the network is the sequence of these computations across all layers, $F_1, F_2, \ldots, F_d$.

Figure 8(a) provides a visualization of such a multi-layer feature circuit. It takes $m_0$ input features and computes features layer by layer, where the computation of a feature $f_{i,j}$ depends only on a subset of features from $F_{i-1}$. For instance, in our earlier simple 2-layer network example (a fully connected layer followed by a single neuron computing a DNF of 2-ANDs), the feature circuit might resemble Figure 9. Note that in that figure, the rectangular portion of the matrix depicts the dependencies (using filled in rectangles), and the formulas to the left of the rectangle depict the actual functions being computed.

This abstract feature circuit finds a concrete implementation in a standard MLP, depicted in Figure 8(b). The $m_0$ input features are first mapped by an embedding matrix $C_0$ (size $n_0 \times m_0$) to an $n_0$-dimensional vector. Subsequent layers $i = 1, \ldots, d$ consist of a weight matrix $W_i$ (size $n_i \times n_{i-1}$), a bias vector $b_i$, and a non-linearity (e.g., ReLU), transforming the $n_{i-1}$ neuron activations from layer $i - 1$ into $n_i$ neuron activations for layer $i$. (While we use ReLU for illustration, other operations like batch normalization could be used instead). A final readout function (e.g., softmax) produces the network's output. Through training (e.g., via gradient descent), this MLP learns to approximate the underlying feature circuit computation.

We will say that a layer $i$ of a neural network *computes in superposition* if $n_i < m_i$, that is, there are fewer neurons than output features in the circuit being computed by this layer. The network as a whole will be

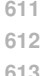
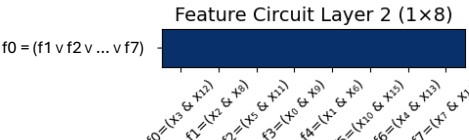
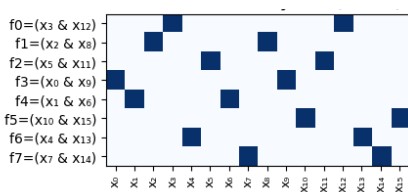

Figure 9: Two layer feature circuit (as an example of the top row of Figure 8) of the DNF $(x_3 \land x_{12}) \lor (x_2 \land x_8) \lor (x_5 \land x_{11}) \lor (x_0 \land x_9) \lor (x_1 \land x_6) \lor (x_{10} \land x_{15}) \lor (x_4 \land x_{13}) \lor (x_7 \land x_{14})$. Layer 1 defines the computation of the pairwise AND features and Layer 2 is the single OR feature over the results of the outputs from Layer 1. Together they define the abstract computation that the network in Figure 10 computes.

said to compute in superposition if its layers compute in superposition. Such networks are sometimes called *polysemantic* because each neuron participates in the computation of more than one output feature, as opposed to monosemantic ones, in which each neuron corresponds to a single output feature.

As Adler & Shavit (2024); Vaintrob et al. (2024) show, a neural network can use a polysemantic representation with many fewer neurons than features to compute across multi-level neural networks. Part of our observations in this work suggest that polysemanticity can arise in training even when the network does not need to be in superposition, that is, $n_i \geq m_i$ (similar to results shown in Lecomte et al. (2024)). In some neural networks, such as large language models or convolutional neural networks, the input to the neural network is provided in superposition. In these cases, the input is encoded into superposition via a first layer that translates text (via tokens) into an embedding space or downsamples an image.

### E.2 THE DISENTANGLED FEATURE CIRCUIT VIEW

Our core proposal for interpreting MLPs involves viewing each layer's weight matrix $W_i$ as implicitly factoring into two conceptual matrices, $C_i$ and $D_i$, revealing an intermediate "disentangled" representation, shown in Figure 8(c). This perspective aims to bridge the gap between the concrete MLP and the abstract feature circuit.

Specifically, we hypothesize that the computation within layer $i$ can be understood as:

1. **Decompression** ($D_i$): An implicit matrix $D_i$ (size $m_{i-1} \times n_{i-1}$) acts on the $n_{i-1}$ neuron activations from layer $i-1$. Its role is to transform the $m_{i-1}$ polysemantic features computed by the previous layer ($F_{i-1}$) into approximations of their monosemantic representations.

2. **Computation and Compression** ($C_i$): An implicit matrix $C_i$ (size $n_i \times m_{i-1}$), combined with the bias $b_i$ and non-linearity $\sigma_i$ (e.g., ReLU), acts on the monosemantic representations. It performs two roles simultaneously, using feature channel coding:
   - It computes the $m_i$ new features for the current layer ($F_i$) based on the $m_{i-1}$ input features.
   - It *compresses* these $m_i$ computed features back into the $n_i$-dimensional polysemantic representations for the next layer.

Thus, the effective transformation by the weight matrix $W_i$ in the standard MLP (Figure 8(b)) is conceptually equivalent to the combined operation $W_i \approx C_i D_i$ in our disentangled view (Figure 8(c)). The entire network can then be viewed as a sequence

$$C_d D_d \sigma_{d-1} \ldots \sigma_1 C_1 D_1 (C_0 \cdot \text{input}),$$

where $\sigma_i$ represents the bias and non-linearity applied after the $C_i D_i$ (or equivalently, $W_i$) multiplication. The actual MLP uses the compact $W_i = C_i D_i$ form, hiding the intermediate dimension of size equal to the

number of features at layer $i$. The names $D_i$ (Decompress) and $C_i$ (Compress) reflect their roles in moving between the compact polysemantic representation and the potentially larger monosemantic representation.

This perspective is similar to concepts like sparse autoencoders and transcoders, where encode/decode operations learn similar transformations. Specifically, the $D_i$ matrix resembles an encoder and the $C_i$ matrix a decoder of a transcoder. However, a key distinction is that $C_i$ and $D_i$ are posited as *intrinsic* components of the *existing* trained weight matrix $W_i$, not externally added modules learned via a separate objective like in standard autoencoders or transcoders. Our goal is to *recover* this inherent $C_i D_i$ structure. Consequently, the placement of non-linearities and biases in our model follows the original network architecture, differing from typical autoencoder/transcoder setups.

Interestingly, if one ignores bias and ReLu, then notice that the feature circuit for any layer $i$ could be given by $D_i * C_{i-1}$, a matrix of size $m_i \times m_{i-1}$ in which the codes of length $n_i$ are matched between $D_i$ and $C_{i-1}$ yielding the layer $i$ feature circuit matrix. So for example, as depicted in Figure 8, the Layer 1 feature circuit would be given by multiplying $D_2$ by $C_1$. This matrix is only an approximation because of the bias and ReLU, whereas $C_1 D_1$ is an accurate representation of the neural network matrix $W_1$.

### E.3  How the Feature Channels Code Computation

Let us spend a moment revisiting how feature channel coding defines computation in this view of the neural network. We assume that $f_{i,j}$, a feature to be computed at layer $i$, is a function of a small set of $k$ features $G = \{f_{i-1,1} \ldots f_{i-1,k}\}$ from layer $i-1$. The features in $G$ of the feature circuit view are each coded (n the neural network view) by a set of neurons from layer $i-1$, and $f_{i,j}$ is coded by a set of neurons from layer $i$. To perform the computation for $f_{i,j}$, the weight matrix $W_i$ maps each of the codes for features in $G$ to the code for $f_{i,j}$. When a feature at layer $i-1$ is used for multiple features at layer $i$, it gets mapped to each of the codes for the features it is used in. Thus, prior to the non-linearity for the activation at layer $i$, each of the inputs to $f_{i,j}$ have been mapped to the same set of neurons at layer $i$. This allows the network to use the code for $f_{i,j}$ as a computational channel for computing $f_0$.

The entangled matrix $W_i$ performs this mapping for the features used by $f_{i,j}$. The disentangled view makes this clear: $D_i$ first maps those features to their monosemantic representation. $C_i$ then brings them all together to the code for $f_{i,j}$. The matrix $C_i$, combined with the bias and the non-linearity, performs the computation required for $f_{i,j}$. In the examples provided in earlier sections, the inputs were already in their monosemantic representation, so there was no need for a $D$ matrix; the layer one weight matrix serves directly as the $C$ matrix as well. Note that the channel's code is somewhat independent of the computation being performed - the goal of channel coding is to bring together the necessary inputs onto the same computational channel. Once they are on the same channel, a number of different functions can be performed on those incoming features. Thus, uncovering the codes for a given neural network would allow us to describe its feature circuit.

### E.4  Cascading Feature Disentanglement

We propose the *cascading feature disentanglement* technique as a potential avenue for recovering the feature circuit from a trained MLP. The core hypothesis is that we can iteratively uncover the matrices $C_i$ and $D_i$ layer by layer.

**Cascading Disentanglement Technique:** For a $d$-layer MLP with weights $W_1, \ldots, W_d$ and input embedding $C_0$:

1. **Initialization (Layer 1):** Estimate $C_1$. Since there is typically no non-linearity between the input embedding $C_0$ (size $n_0 \times m_0$) and the first weight matrix $W_1$ (size $n_1 \times n_0$), the initial disentanglement is simply $C_1 \approx W_1 C_0$. The role of $D_1$ (size $m_0 \times n_0$) is to invert the input embedding, i.e., $C_0 = D_1^*$ (where $D_1^*$ denotes an approximate right inverse of $D_1$, as $D_1$ is generally

non-square Wikipedia contributors (2025)). If this holds, then since $W_1 \approx C_1 D_1$, we see that
$W_1 C_0 \approx C_1 D_1 C_0 = C_1 D_1 D_1^* \approx C_1$.

2. **Iteration (Layer $i > 1$):** Assume we have successfully estimated $C_{i-1}$. Our goal is to estimate $C_i$ from $W_i \approx C_i D_i$. Unlike Step 1, we cannot simply assume $D_i \approx C_{i-1}^*$ because a non-linearity and bias ($\sigma_{i-1}$) were applied after the computation involving $C_{i-1}$. The central hypothesis for this step is: By analyzing the estimated $C_{i-1}$ (size $n_{i-1} \times m_{i-2}$), we can potentially deduce the $m_{i-1}$ features computed by layer $i-1$ and how they are represented by the $n_{i-1}$ neuron activations (after bias and ReLU). This understanding of the layer $i-1$ output features could allow us to construct an estimate for $D_i$ (and hence its approximate inverse $D_i^*$). If we can find $D_i^*$, we can then estimate $C_i \approx W_i D_i^*$. Repeating this process for $i = 2, \ldots, d$ would, in principle, decode the entire network's feature circuit via $C_1, \ldots, C_d$.

Developing the precise methods to infer the layer $i-1$ feature vectors from $C_{i-1}$ and construct the corresponding $D_i$ (or $D_i^+$) remains a significant challenge and is a key direction for future research. This cascading approach, however, provides a conceptual roadmap for systematically disentangling deep MLPs.[6] We also point out that an alternative approach would be to use a sparse autoencoder to determine the feature encoding being input to layer $i$, and use that to construct the pseudo-inverse $D_i^+$. In fact, the decode matrix of the sparse autoencoder may itself serve as the pseudo-inverse $D_i^+$, since it essentially performs the same function for layer $i$ as an initial embedding matrix does for layer 1.

### E.5 CASCADING FEATURE DISENTANGLEMENT WITH AN EMBEDDING LAYER

We here demonstrate that that Cascading Disentanglement works for a small example network where we add an embedding layer to the two layer networks we have described earlier in the paper. Specifically, Figure 10 depicts three example networks trained via gradient descent, where three different types of embeddings are added to the feature circuit depicted in Figure 9. Each row of matrices is a different type of embedding, where the rightmost column depicts the embedding being used: in the middle row, the embedding $C_0$ is the identity function (i.e., no embedding), in the top row it is a Hadamard matrix, and the bottom row it is a symmetric random embedding.

The middle column depicts the raw matrix $W_1$, i.e., prior to disentanglement. In the case of the identity embedding (middle row) feature channel coding is clear. In the case of the Hadamard and random embeddings, the $W_1$ matrices look random. The leftmost column depicts the result of disentanglement: the matrix obtained by the product $W_1 C_0$. As described above, this gives us $C_1$ since $W_1 C_0 \approx C_1 D_1 C_0 \approx C_1 D_1 D_1^* \approx C_1$. This reveals the feature channel encoding hidden within $W_1$ for both the Hadamard embedding and the random embedding. In other words, multiplying by $C_0$ undoes the implicit embedding that was learned by the network in $W_1$, and reveals the feature channel codes.

This is a good place to pause and contemplate the power of disentanglement. The relative magnitudes of the weights in the raw $W_1$ matrices, before disentanglement, might be quite misleading. For example, in the Hadamard embedding example, it would seem from $W_1$ that for Neuron 3, the weights in columns 2 and 15 are the most important (as they are of the largest magnitude), and yet we know from the disentangled matrix that these weights are just the result of the embedding space: the weights in these columns do not explainwhat neuron 3 is contributing to the computation. From the disentangled matrix $C_1$ we can tell that Neuron 3 is part of the coding for fetaures ($x_2 \wedge x_{12}$) and ($x_0 \wedge x_9$). This is the power of combinatorial disentanglement: If we want to decide which weights or neurons to prune, which to apply dropout to during training and so on,

---

[6]Note that we are proposing an algorithmic framework without having the solution algorithm, apart from some initial heuristic attempts. Skeptics might criticize us for this. We note that this has been done before, for example, in the context of public key cryptography Diffie & Hellman (1976) where Diffie and Hellman proposed a public key cryptography algorithm without the RSA algorithm Rivest et al. (1978) to implement it.

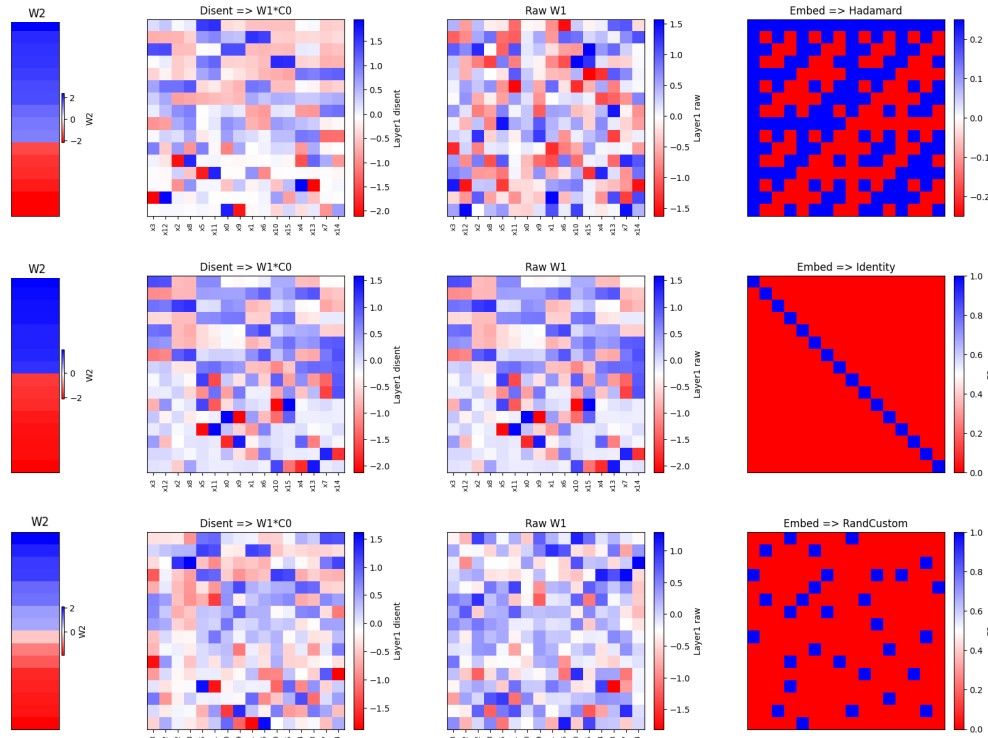

Figure 10: Disentangling of networks computing the feature circuit depicted in Figure 9. In top a 2-layer network with an initial Hadamard embedding layer $C_0$ that is trained as before on a DNF of 8 different 2-AND clauses. The raw trained $W_1$ matrix is shown immediately to the left of it. Then to its left we show the disentangled $C_1 = W_1 C_0$. Notice that $C_1$ has the usual expected structure with coding of 2P in the positive witness rows and coding for a XOR using 1N1P in the negative witness rows. Moreover, though having less 2P patterns and positive rows, it has a similar structure as the case where there is no embedding (the embedding is the identity matrix) as shown in the middle row of the figure. The bottom row shows how a symmetric random embedding is disentangled in a similar fashion.

doing so in the disentangled space would give us the insights that are missing when using activation space approaches.

In the above experiments we have used three different types of embeddings: identity (or no embedding), using a Hadamard matrix, and using a symmetric random embedding. In most of the statistics in Section C we trained on a network where the input had a Hadamard embedding layer $C_0$ which we then used to get $W_1 * C_0$, the coding matrix in which we counted the number of patterns (as a sanity check, we also ran the same training without the Hadamard embedding, getting similar results). In all other sections we used the identity embedding.

Finally, one can understand today's prevailing approach to interpreability via the training of an autoencoder Rajamanoharan et al. (2024a) using our coding-decoding view of computation. The autoencoder

approach is a way of learning an approximation of $D_i$ for a given layer $i$. The autoencoder has a very large inner layer to which the output activations of $C_{i-1}$ are mapped, along with a sparsity constraint to separate them so that the features of layer $i - 1$ can be read. The difference in the combinatorial approach we propose here is to attempt to decode $C_{i-1}$ and reconstruct $D_i$ by analyzing the network weight matrices themselves rather then learning features from the activations using a separate trained network. As mentioned above, we think there is potential in a combined approach, for example where $D_i^+$ is learned via an autoencoder (or other activation based learning approaches Dunefsky et al. (2024b)), and then this is used to find and interpret $C_i$ combinatorially. This would allow our approach to be applied to a single layer, without the need for cascading.

## F  TOWARDS DEEPER NETWORKS

As a bit of a teaser as to the potential directions one can follow with the combinatorial interpretability approach, we examine what happens if there are multiple outputs - i.e., a classification problem with more than 2 possibilities. To do so, we add additional Boolean formulas to learn, and for each additional formula, there is a corresponding additional second layer output neuron responsible for that formula. The inputs and the first hidden layer are shared between the different outputs, and so the output neurons are computing different formulas on the same set of Boolean variables and on the same hidden layer results. We still only consider the case where there is a single hidden layer, but we view increasing the size of the output layer as a small step towards having an additional hidden layer, as view computing multiple output features as a potential proxy for computing multiple features for use at the next hidden layer of the network.

We first consider the case with two output neurons. We tested a range of clause numbers, and found something similar to the single output case: the system is able to train fairly successfully, provided that the total number of clauses (in this case, summed across all outputs) is not larger than the hidden dimension. We here provide results for one representative example: in this case there are 32 Boolean variables and a hidden dimension of 32. The formulas are in DNF form with clauses of size 4 (4-AND), and each output is the OR of 8 clauses. The clauses for each output are non-overlapping in terms of variables, but the clauses for one output are chosen without taking into account the other output. Thus, each variable is used exactly once in each output but can be used in both outputs.

We generated 50000 inputs, each having between 8 and 10 variables set to true. The inputs were constructed independently, and each had equal likelihood of being each of the 4 output combinations (00, 01, 10, 11). Each output possibility was constructed in the same style as used in Section C.

We provide the full neural network parameterization for the following pair of formulas:

$$f_0(x) = (x_3 \wedge x_4 \wedge x_7 \wedge x_{10}) \vee (x_2 \wedge x_6 \wedge x_{25} \wedge x_{27}) \vee (x_0 \wedge x_{13} \wedge x_{30} \wedge x_{31}) \vee$$
$$(x_9 \wedge x_{16} \wedge x_{20} \wedge x_{21}) \vee (x_8 \wedge x_{12} \wedge x_{14} \wedge x_{28}) \vee (x_1 \wedge x_5 \wedge x_{17} \wedge x_{24}) \vee$$
$$(x_{11} \wedge x_{22} \wedge x_{26} \wedge x_{29}) \vee (x_{15} \wedge x_{18} \wedge x_{19} \wedge x_{23})$$

$$f_1(x) = (x_{10} \wedge x_{15} \wedge x_{23} \wedge x_{26}) \vee (x_9 \wedge x_{13} \wedge x_{18} \wedge x_{19}) \vee (x_0 \wedge x_4 \wedge x_8 \wedge x_{14}) \vee$$
$$(x_6 \wedge x_{20} \wedge x_{28} \wedge x_{31}) \vee (x_3 \wedge x_{21} \wedge x_{22} \wedge x_{30}) \vee (x_2 \wedge x_{11} \wedge x_{12} \wedge x_{24}) \vee$$
$$(x_1 \wedge x_5 \wedge x_{25} \wedge x_{29}) \vee (x_7 \wedge x_{16} \wedge x_{17} \wedge x_{27})$$

This computation is depicted in Figure 11. This network achieved training loss of 3.42% after 1000 epochs, and at that time had overall accuracy (both bits correct) on the test set of 97.30%. Not only did the network

learn this network very clearly, we see clear fature channel coding, despite having to do that coding for two different Boolean functions simultaneously.

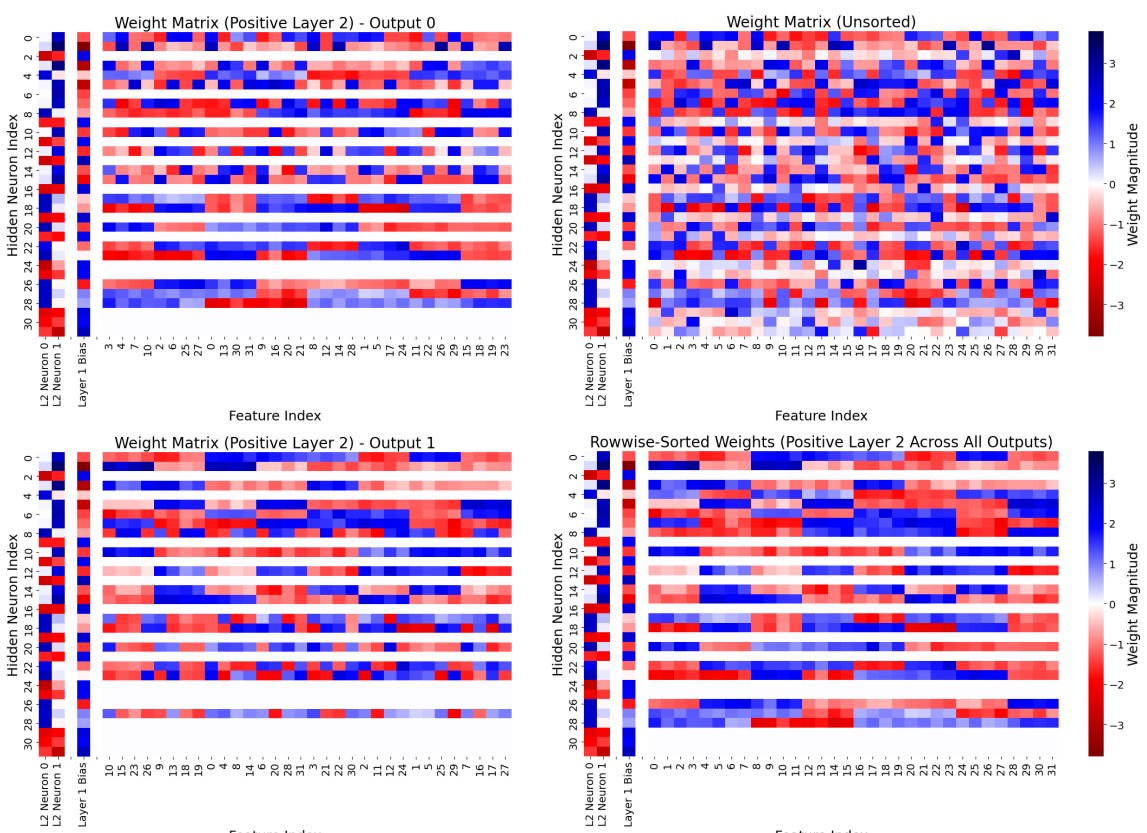

Figure 11: Trained neural network for $f_0$ and $f_1$. The top right is the unsorted version of the weight matrix, with all weights and biases. The top left is sorted by the clauses of $f_0$ (computed by output 0), and, as above, filtered to show only those neurons that have a positive edge to output neuron 0. The bottom left is the equivalent version for output neuron 1 and $f_1$. The bottom right has all rows with a positive edge to either output neuron, and each such row is sorted by the clauses corresponding to the output neuron with the larger weight to that row/hidden layer neuron.

A few observations about this parameterization:

- The biases are right in line with our expectations: if *either* output neuron has a positive weight to a hidden layer neuron, then the bias for the hidden layer neuron is negative. (Neurons 27 and 28 are exceptions and slightly positive.) All rows where both output layer weights are negative have a positive hidden layer bias.

- The positive row filtering for a given neuron sometimes codes exactly as expected, and other times it seems random. However, on closer examination, in every single case where it seems random, the weight from the other output neuron to that row is significantly larger. It is really coding for the other neuron in those rows, and simply looks random because we are sorting by the wrong set of clauses.

- The output neuron weights partition the hidden layer neurons into three categories: (1) positive and larger weight from output neuron 0, (2) positive and larger weight from output neuron 1, and (3) negative from both neurons. Neurons of type (1) code for output neuron 0 and function $f_0$. Neurons of type (2) code for neuron 1. Neurons of type (3) prove negative witnesses. This is visualized clearly in the bottom right panel of Figure 11. There, each row is sorted according to which cell of the partition is in: for rows of type (1), we sort by the clauses of $f_0$, for rows of type (2), we sort by the clauses of $f_1$, and rows of type (3) are filtered out. We see that the result is nearly perfect coding.

- Once the system has chosen which output a row is coding for, it effectively ignores the other output for that row - the other weight has a much smaller magnitude weight, and that weight has a tendancy to be (slightly) positive.

- In Section C, we saw that even as we increased the number of clauses, the network was not able to learn a weight distribution with more than roughly half positive rows. We here see a fairly even divide between the three cells of the partition, and so the number of negative rows is only 12. This is additional evidence that the system does not need as many negative rows as the solution that it is converging to has. For this case where there are 2 output neurons, one could also argue that it is seeking equality between the positive rows and the negative rows, and the negative rows are reused between the two output neurons. However, (a) we do not see that same effect with 3 output neurons (below), and (b) even if that were the case, it does imply a limit on how much logic can be built into the negative rows (since they are the same between the two outputs, with relatively minor differences in how negative the output layer weight is).

We next turn to the case where we have 3 output neurons. The setup is the same otherwise, except that we now train with more data (200,000 inputs). The result was slightly worse performance for this more difficult case (since we now have 24 total clauses, instead of 16). Specifically, after 1000 epochs, we have training loss 5.86% and test set accuracy (all 3 outputs correct) of 95.05%. Figure 12 depicts the results for the following set of formulas:

$$g_0(x) = (x_5 \wedge x_{10} \wedge x_{11} \wedge x_{20}) \vee (x_4 \wedge x_7 \wedge x_{14} \wedge x_{27}) \vee (x_{12} \wedge x_{17} \wedge x_{21} \wedge x_{24}) \vee$$
$$(x_{13} \wedge x_{15} \wedge x_{28} \wedge x_{30}) \vee (x_2 \wedge x_6 \wedge x_9 \wedge x_{22}) \vee (x_8 \wedge x_{23} \wedge x_{29} \wedge x_{31}) \vee$$
$$(x_0 \wedge x_{16} \wedge x_{25} \wedge x_{26}) \vee (x_1 \wedge x_3 \wedge x_{18} \wedge x_{19})$$

$$g_1(x) = (x_0 \wedge x_{10} \wedge x_{19} \wedge x_{27}) \vee (x_{11} \wedge x_{12} \wedge x_{14} \wedge x_{24}) \vee (x_2 \wedge x_3 \wedge x_4 \wedge x_7) \vee$$
$$(x_6 \wedge x_8 \wedge x_9 \wedge x_{20}) \vee (x_{16} \wedge x_{22} \wedge x_{26} \wedge x_{30}) \vee (x_1 \wedge x_{21} \wedge x_{25} \wedge x_{29}) \vee$$
$$(x_{17} \wedge x_{23} \wedge x_{28} \wedge x_{31}) \vee (x_5 \wedge x_{13} \wedge x_{15} \wedge x_{18})$$

$$g_2(x) = (x_1 \wedge x_5 \wedge x_{21} \wedge x_{30}) \vee (x_6 \wedge x_{12} \wedge x_{13} \wedge x_{20}) \vee (x_2 \wedge x_{17} \wedge x_{22} \wedge x_{29}) \vee$$
$$(x_{10} \wedge x_{26} \wedge x_{28} \wedge x_{31}) \vee (x_4 \wedge x_{11} \wedge x_{15} \wedge x_{25}) \vee (x_0 \wedge x_7 \wedge x_{16} \wedge x_{18}) \vee$$
$$(x_3 \wedge x_8 \wedge x_{23} \wedge x_{27}) \vee (x_9 \wedge x_{14} \wedge x_{19} \wedge x_{24})$$

This solution is very similar to the case with 2 output neurons, and as we can see in Figure 12, it is clearly using feature channel coding. However the structure of the network is not quite as clean. This is not surprising, given (a) the poorer performance, and (b) the fact that we are starting to approach the limit we discussed in Section C, where the total number of clauses equals the size of the hidden dimension. Specifically, we

see that the layer 1 biases are not as consistently negative for the positive rows, and that not every positive row chose a clear output to code for (see neuron 27, which just contributes a bit of positive value from all input variables. It is however, interesting to note that the number of negative rows is only 3 (or 4 if you count row 19), leaving each output about 9 rows of positive coding, again providing evidence that the single output solutions are over investing in negative rows.

We did also experiment with 4 output neurons, but our initial attempts at that did not yield particularly effective networks. In that case the number of clauses was equal to the size of the hidden layer; the trained networks tended to have error rates around 1/3. Still even in these poorly performing networks, it was clearly trying to use feature channel coding.

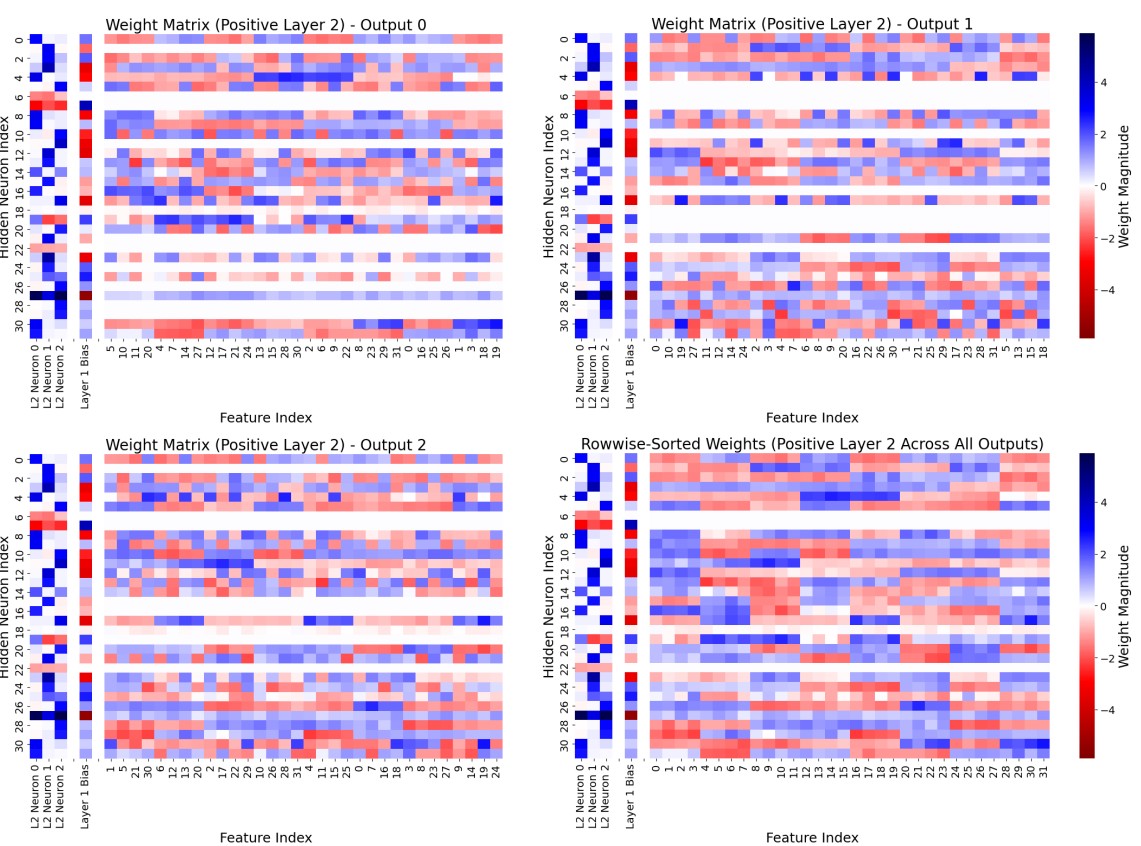

Figure 12: Trained neural network for $g_0$, $g_1$, and $g_2$. The top right is sorted by the clauses of $g_0$ (computed by output 0), and filtered to show only those neurons that have a positive edge to output neuron 0. The top left is the equivalent version for output neuron 1 and $g_1$, and the bottom left is the equivalent for output neuron 2 and $g_2$. The bottom right has all rows with a positive edge to any output neuron, and each such row is sorted by the clauses corresponding to the output neuron with the largest weight to that row/hidden layer neuron.

## G    COMBINATORIAL ANALYSIS OF A SCALING LAW

We here provide significantly more detail on the analysis found in Section C. We first provide additional details on how the networks were trained. For each trial, we train for up to 100 epochs (with patience-based early stopping) on 20,000 samples. The samples are drawn from the following distribution: with 50% probability, we assign a label of 1 (True), randomly select one of the formula's clauses, set its variables to satisfy that clause, and then choose between zero and two additional variables to set to 1 while preserving the clause's satisfiability. Otherwise (label 0, a value of False), we again pick a clause but now deliberately "break" it so that only three of the four variables are set to satisfy that clause. We also again choose additional variables to set to 1, so that the total number of set variables will be between four and six while the value will still be False. We use the Adam optimizer with a learning rate of 0.001, a batch size of 64, and a patience of 10 epochs for early stopping. We note that the distribution matters quite a bit for the overall numbers of patterns we report on, but not the general scaling trends that we describe.

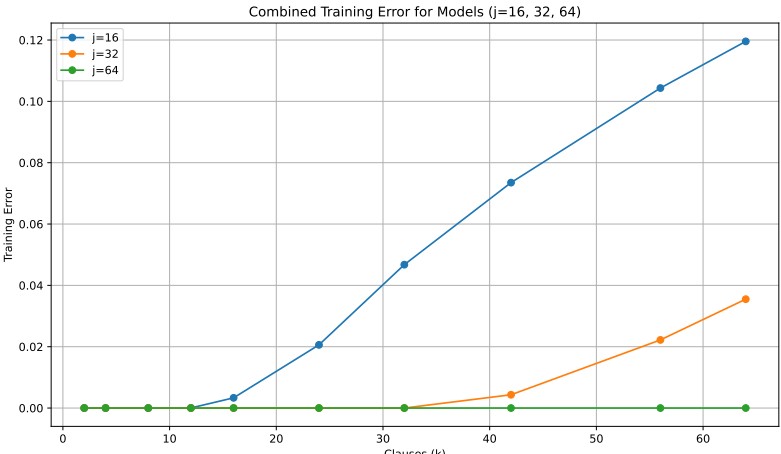

Figure 13: Onset of training error as number of clauses increases.

In Figure 13, we examine training error for the three sizes of the hidden layer for the case of clauses of four non-negated variables, and see that it starts to accumulate when the number of clauses reaches the number of neurons in the hidden layer. Figure 14 provides all three versions of the scaling graphs presented in Figure 6. For ease of comparison, we provide the $j = 32$ case here again, along with the $j = 16$ and $j = 64$ cases not presented in Section C. For an explanation of these graphs, see Section C. Each point in Figure 14 is the average of 10 different runs of model training. We also provide the full distribution of these samples for the 4P case in Figure 15. We there observe that for some values of $j$ and $k$ there is significant variability in how many coding rows appear. However, the more constrained the capacity of the Layer 1 weight matrix is, the tighter that variance is, and the more concentrated those values are around the upper limit of that capacity.

Next, in the three versions of Figure 16, we graph the percentage of hidden layer neurons that are connected to the output via positive weights. As we saw in Figure 1, the sign of that weight differentiates how the model uses that neuron, for positive or negative witnesses. We see that in these tests, the learned model does not stray far from having half of the neurons in positive rows and half in negative rows. In other experiments, not described in this paper, we see that the fraction of negative and positive rows is strongly influenced by the fraction of positive and negative examples in the training set. For example, if the training set has 90% positive examples instead of 50% positive examples, there will be very few negative rows. This is perhaps not

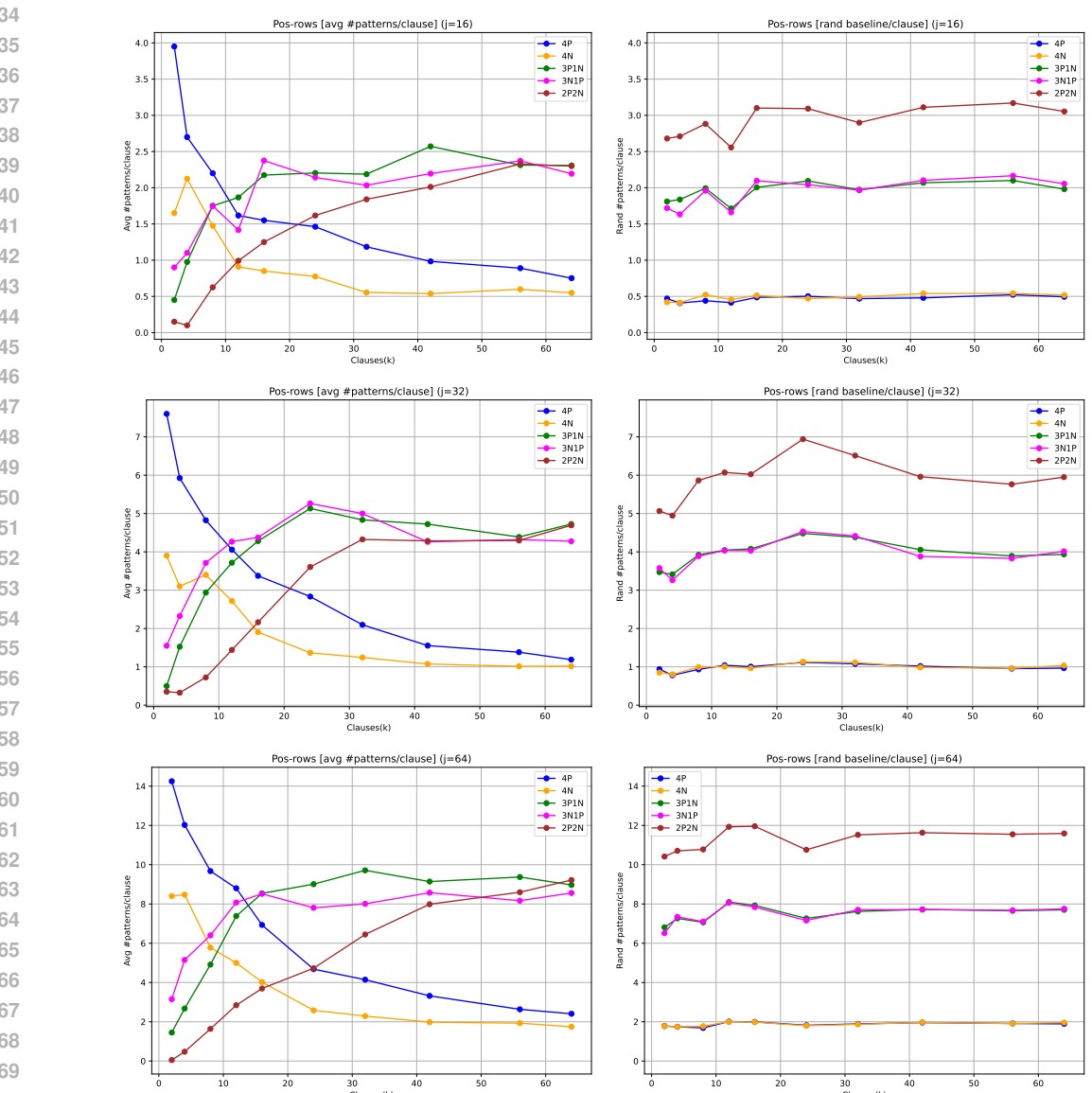

Figure 14: Prevalence of coding patterns in positive rows: trained networks versus random networks.

that surprising if viewed through the lens of gradient descent: positive examples can only either increase or not change Layer 2 weights of our network, and negative examples can only decrease or not change Layer 2 weights. Fully understanding and quantifying this phenomenon is an interesting open problem, but for this work, we maintained a balanced distribution of positive and negative examples.

Figure 16 also depicts how many of the positive rows also have a negative bias (which is what feature channel coding would predict). Interestingly, we see that for the (simpler) functions with a smaller number of clauses,

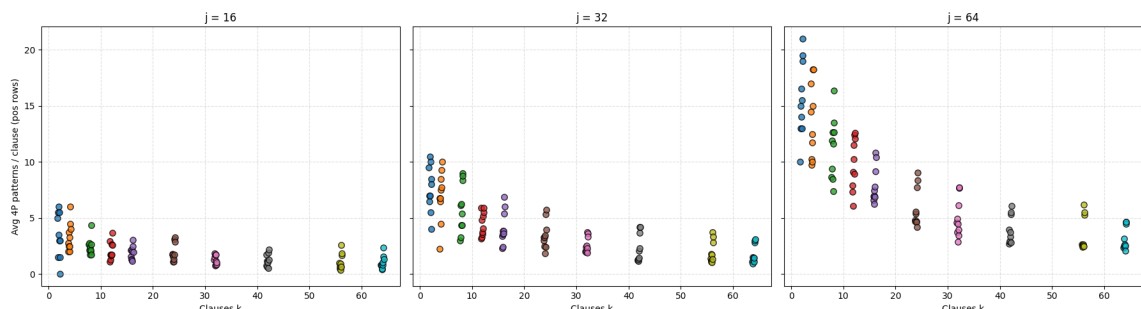

Figure 15: The distribution of 4P coding patterns observed in positive rows of the Layer 1 weight matrixes. Each point is one of 10 models for each value of $j$ and $k$; their average value is depicted in Figure 14. As can be seen, when there is sufficient room for multiple coding rows per clause, there is some variability on the size of the codes, but as code capacity becomes a constraint, the distribution becomes more concentrated.

the system does indeed have all (or almost all) of the positive Layer 1 neurons associated with a negative bias. However, as the function becomes more complex, more and more of the second layer neurons have positive bias. And this does not line up with the point where the network training accuracy falls off - it happens around 15 to 20 clauses for all three values of $j$, and so seems independent of the size of the hidden layer. It is also past the point where accuracy falls off for 16 hidden neurons, but before it for the two cases of a larger hidden layer, and so it seems that having negative bias is helpful, but not crucial to learning the Boolean function. We will discuss below a possible reason for why the percentage of positive rows with negative bias decreases the way it does.

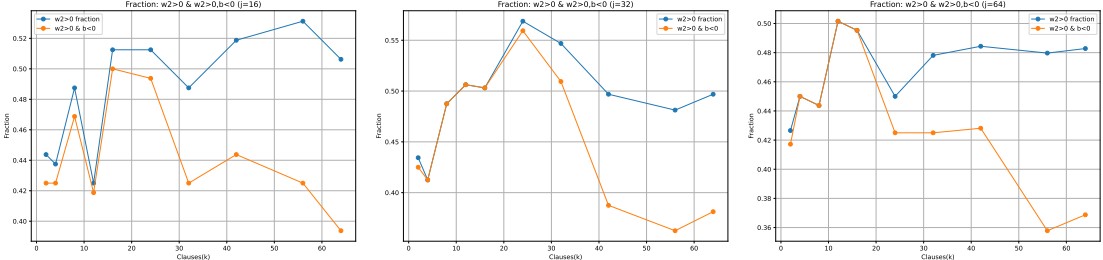

Figure 16: Fraction of positive Layer 2 weights and negative Layer 1 bias.

We next turn to the three versions of Figure 17. This depicts the fraction of entries of the hidden layer weight matrix that are positive and negative. We examine the overall number, as well as the fraction in positive rows that are in inputs used in at least one clause, positive rows that are in inputs not used by any clauses, and similarly for negative rows. We see that overall, very close to half of the matrix entries are positive, and that there is a small positive bias for the positive rows and a small negative bias for the negative rows. This small bias decreases as the number of clauses increases. We also see that the non-clause variables have the opposite bias of the clause variables, although closer examination (not depicted in the graph) reveals that the magnitude of these weights is much smaller than that of the clause variables, and so we do not believe this impacts how the neural network learns very much. Both the fact that there is so little bias in the clause variables and the fact that the three graphs are so similar (with no discernable difference where training error occurs) leads us to believe that it is useful for the network to have close to an even mix of positive and negative values, but this measure does not explain training error.

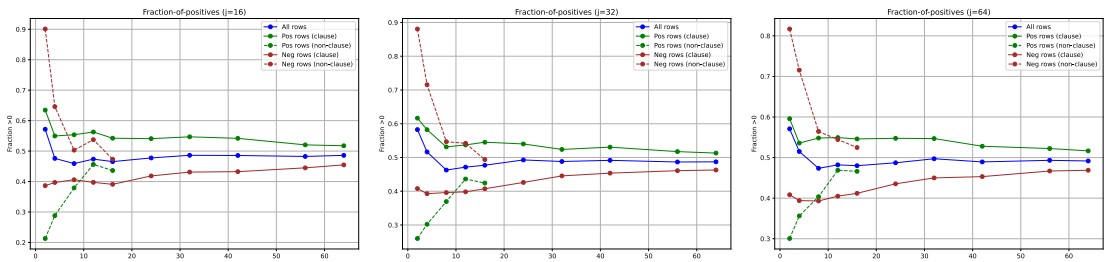

Figure 17: Bias of Layer 1 matrix weights, overall and by row and variable type

### G.1 THEORETICAL LIMITS OF THE 4P CODING PATTERN

We next dive deeper into the limit on how many 4P patterns can be packed into the weight matrix, described in Section C. As pointed out there, the saturation of the 4P patterns seems to be a real limitation of the network. We first point out that the positive rows have roughly half positive (coding) values, and half of them are negative. This is necessary to group the inputs contained in a feature together - if all inputs were positive it would not be possible to distinguish 4 inputs from the same AND clause being true from 4 inputs from different AND clauses being true. Thus, we assume here that if we have $\ell$ input variables, each row will have roughly $\ell/2$ positive values that can go towards 4P patterns. Let $\rho$ be the fraction of positive rows. If there are $j$ hidden neurons, then the total number of positive entries in positive rows will be roughly $\frac{j\ell\rho}{2}$.

Each 4P pattern requires 4 of those positive entries. There may be some overlap between the 4 positive entries of different 4P patterns, but since the clauses are chosen randomly, that overlap will be small, and we shall ignore that effect here (but we note that for clauses of size 2 that overlap is much more significant, and in fact we see that networks trained on such smaller clauses can handle a much larger number of clauses). As a result, the average number of 4P patterns per clause, when there are $k$ clauses, will be $\frac{j\ell\rho}{8k}$. Since $\rho$ is approximately $1/2$, this will be roughly $\frac{j\ell}{16k}$ This explains the saturation point we see in the above graphs. For example, when $j = k = \ell = 32$, according to this formula, we would not expect to see more than 2 occurrences of 4P per clause. Comparing this prediction to Figure 6, we see that our prediction is accurate: the value of the blue curve in the $j = 32$ graph of the right hand column, at the 32 clause point is almost exactly 2. In other words, the network in practice was able to achieve this maximum packing of 4P patterns, but not better.

If we increase $k$ to 64 and hold the other variables fixed, our formula says there is not room for more than an average of one 4P per clause; again Figure 6 shows us the network is able to achieve that but not much higher (we suspect the slight overperformance is the result of increasing overlap of clauses). Conversely, when we decrease $k$, the network does not quite keep up with the theoretical maximum. But in those cases, the network does not have an incentive to keep up with that maximum: there is plenty of room for coding and so it does not have to saturate the network. We see a similar agreement with this packing limit for the $j = 16$ and $j = 64$ curves for 4P in Figure 6. In all cases, when the network is no longer able to achieve an average of approximately 2 coding rows per clause, it starts to accumulate error.

### G.2 CODES AS NEGATIVE WITNESSES

We next turn our attention to the negative rows of the weight matrix, as depicted in Figure 18. These graphs are analogous to those in Figure 6, but focus on patterns emerging in negative rows, again using a random matrix baseline configured with the corresponding row counts and biases. For a small number of clauses, the 3N1P pattern exhibits the strongest signal in these negative rows. However, as the number of clauses

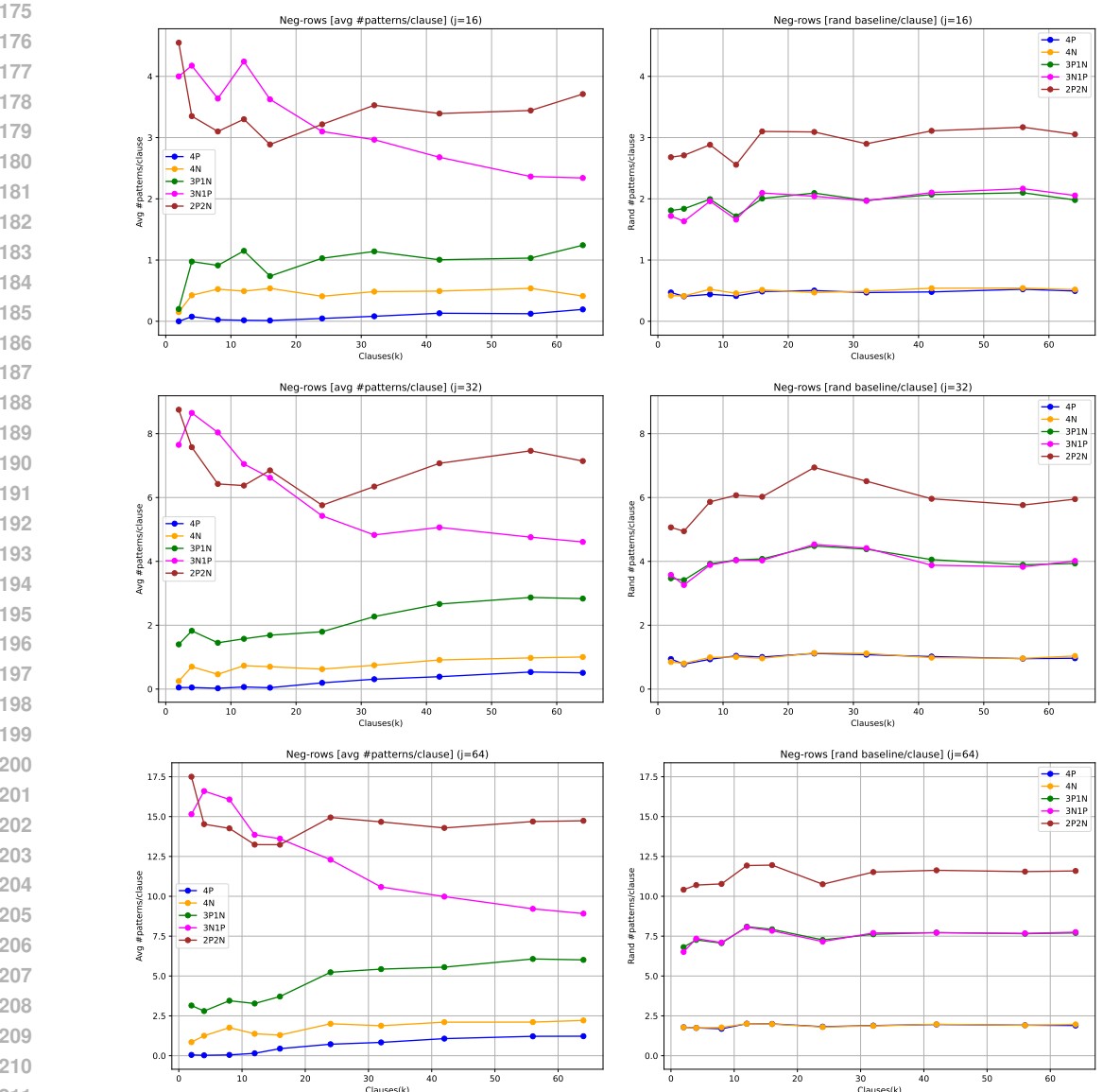

Figure 18: Prevalence of coding patterns in negative rows: trained networks versus random networks.

increases, the dominant pattern transitions to 2P2N. Both patterns appear designed to detect instances where two or three positive variables, but not all four, are present in a clause. This detection generates a positive post-activation signal at Layer 1, which is then inverted to a negative signal by the negative Layer 2 weight. The 3N1P pattern achieves this using a positive Layer 1 bias, roughly equal negative Layer 1 weights, and ensuring the sum of the bias and the positive weight approximately offsets the sum of the three negative weights (e.g., weights: -1, -1, -1, +2; bias: +1). In this configuration, if the variable corresponding to the positive pattern column is 1, all three variables corresponding to negative columns must also be 1 to suppress

a positive post-activation value. However, if any of the three negative weight variables are 0 (indicating the clause condition is not met), a positive Layer 1 activation results, leading to a negative final signal via Layer 2. As the clause count increases, there is no longer sufficient room to fully utilize the 3N1P pattern, and the mechanism shifts towards the 2P2N pattern. This coincides with a reduced reliance on positive bias in the negative rows. Additionally, the 3P1N, 4P, and 4N patterns are consistently suppressed in negative rows, indicating they are not effective for feature channel coding to be performed by these rows.

## G.3 CODE INTERFERENCE

As described above, the number of 4P coding rows a clause coincides with seems to have a significant impact on the network's ability to learn the Boolean function. We take this a step further in Figure 19, where we depict the number of clauses that do not coincide with any 4P rows. This is depicted on the right side - the left side is a duplicate of Figure 13, shown again here for convenience. We see here that there is, in fact, a close alignment between training error and the number of clauses with no 4P coding rows. However, it does seem like the network can tolerate a small amount of clauses without 4P coding rows.

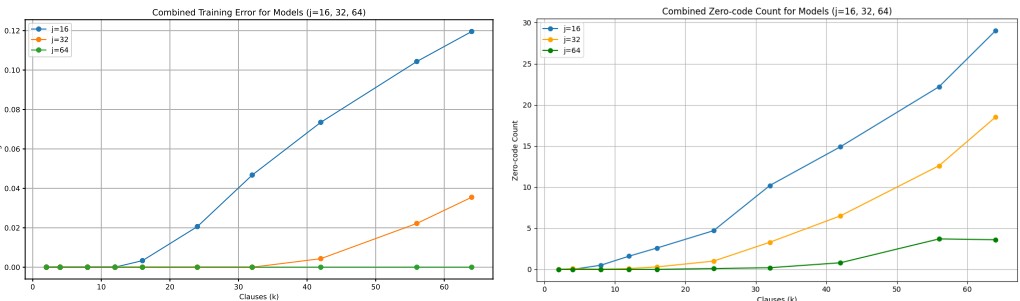

Figure 19: Scaling of training error and number of clauses without codes

Next, we examine the overlap between the codes that the network is using. As shown in Adler & Shavit (2024), this is a central aspect of coding. Codes usually overlap and always will if there is sufficient saturation of the network by the number of features being coded. However, as long as the overlap between pairs of codes is not too large, this is fine. One of the main advantages of using codes is that spreading out a signal across multiple neurons creates tolerance to noise that appears on some of those neurons. This is a core tenant of the feature channel coding hypothesis.

We show the code overlap measured from our experiments in Figure 20. We plot the average, over all pairs of clauses in each training run, of the overlap between the codes, where an overlap means that both codes use the same row as a 4P positive row. We also show the total number of 4P coding rows. For both values, we do not include clauses without coding rows in the count, so the blue curve (code size) will differ from that of Figure 6, which includes those clauses. These curves are almost identical (except for scaling) for the three hidden parameter sizes, and consistent with what we would expect from the feature channel coding hypothesis. Also, we do not see any indication that this overlap is responsible for the network not being able to train.

## G.4 CLAUSES WITH A NEGATED VARIABLE

So far we have shown how feature coding appears in DNFs that have all positive variables. In this section we show how features with negative variables are coded, by studying DNF clauses with 3 non-negated variables and 1 negated variable. We will see that with this slightly more complex formula, the overall coding behavior

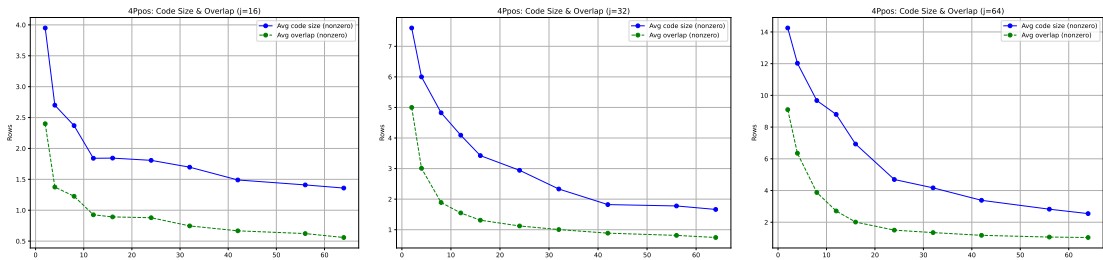

Figure 20: The average, over all pairs of clauses in each training run, of the overlap between the feature channel codes. Average code size is shown to give the reader a measure of the relative portion of the overlaps.

is very similar to the fully positive case. We also use this example to show how computation with feature channel codes proceeds.

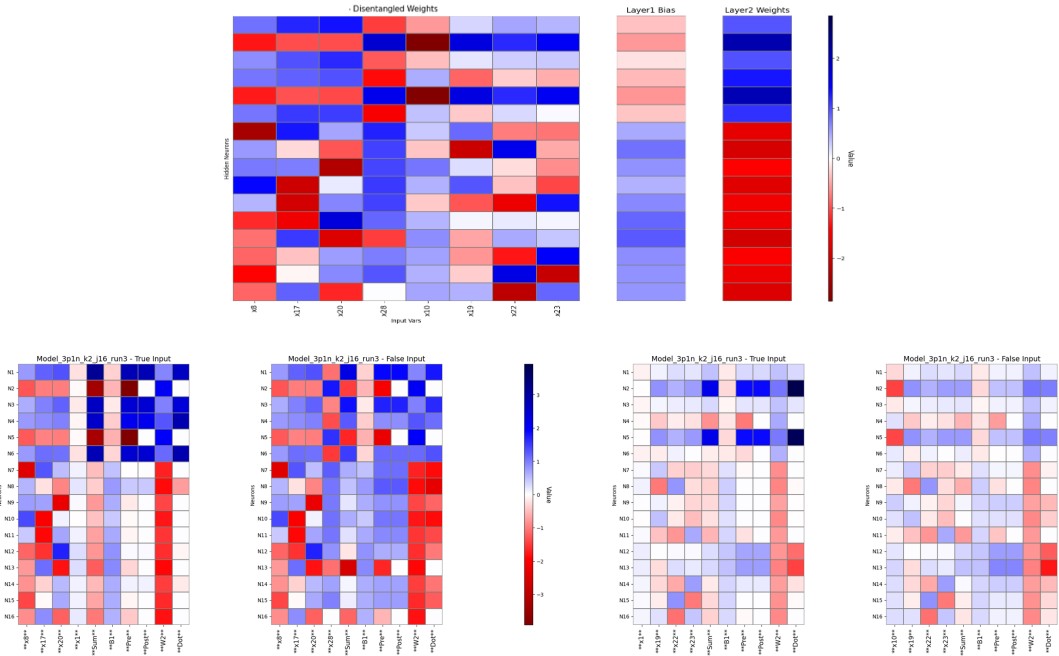

Figure 21: Execution traces show how channel coding in a model with clauses that have a negated variable each deliver the appropriate positive or negative response.

We trained the same simple network with a single hidden layer as in our prior experiments, here with 32 input Boolean variables and 16 hidden neurons. In the top part of Figure 21, we show the result of training the model with the formula $(x_8 \wedge x_{17} \wedge x_{20} \wedge \neg x_{28}) \vee (\neg x_{10} \wedge x_{19} \wedge x_{22} \wedge x_{23})$. For legibility we are only showing the columns corresponding to the 8 variables in the 2 clauses, not all 32 of them. The input features are sorted in the order of the clauses. A quick glace at the matrix shows us the feature channel coding patterns. As before, the network has converged to have a negative bias in Layer 1 neurons corresponding to positive

witnesses (positive Layer 2 weights), and positive Layer 1 bias in neurons corresponding to negative witnesses (negative Layer 2 weights). Looking at the Layer 1 matrix entries for the clause $(x_8 \wedge x_{17} \wedge x_{20} \wedge \neg x_{28})$, we see clear positive coding rows emerge in neurons 1, 3, 4 and 6. For the clause $(\neg x_{10} \wedge x_{19} \wedge x_{22} \wedge x_{23})$, the positive coding rows are 1, 2, 3, and 5. Thus, the positive codes overlap in neurons 1 and 3, but the coding for the second clause has smaller magnitude weights in those overlapping rows (which is not a coincidence - see below.) The Boolean computation performed using these feature channels is as expected: the negated variables have corresponding columns with negative weights while the columns corresponding to positive variables have positive weights. As before we call this pattern 3P1N.

The bottom part of Figure 21 presents four execution traces for the model evaluating two clauses: $(x_8 \wedge x_{17} \wedge x_{20} \wedge \neg x_{28})$ (left two traces) and $(\neg x_{10} \wedge x_{19} \wedge x_{22} \wedge x_{23})$ (right two traces). Each trace details network activation over time (left-to-right), showing initial True variable inputs (first 4 columns), intermediate Layer 1 computations (dot product, bias, pre/post-ReLU values), Layer 2 weights, and the final Layer 1 neuron contributions to the Layer 2 output.

For the first clause, Trace 1 shows a satisfied case ($x_8, x_{17}, x_{20}, x_1$ True and $x_{28}$ (not shown) False). Positive neurons (1, 3, 4, 6) activate according to $x_8, x_{17}, x_{20}$, while negative neurons (7-17) remain mostly inactive, resulting in a correct sigmoid output of 0.9999 (True). Trace 2 shows an unsatisfied case ($x_8, x_{17}, x_{20}, x_{28}$ True). Here, $x_{28}$ inhibits positive neurons (1, 3, 4, 6) via negative weights and activates negative neurons (7-17) via positive weights, resulting in a correct sigmoid output of 0.0015 (False).

For the second clause, Trace 3 demonstrates that in a satisfied case $x_{19}, x_{22}, x_{23}, x_1$ True and $x_{10}$ (not shown) False), positive activation occurs primarily through the set of neurons (2 and 5) that does not overlap with the activations in Trace 1 above. This indicates something impressive: the network has found feature channel coding that mostly separates clause representations. Trace 4 shows an unsatisfied case ($x_{10}, x_{19}, x_{22}, x_{23}$ True). Variable $x_{10}$ suppresses the positive signal in neurons 2 and 5 via negative weights and activates negative witness neurons (9-16) via positive weights, leading to a False output.

As before, we could have hand constructed codes based on two monosemantic neurons, one for each of the clauses, and used large negative biases to provide a clean signal. However, the technique that gradient descent found involves coding using both positive and negative witnesses, where the negative witnesses might be partly playing the role of the bias in the final computation. In the final result, the coding is clear but uses varying weights, not clean binary values as in the hand crafted combinatorial feature channel codes presented in Adler & Shavit (2024), most likely due to the pecularities of gradient descent. We believe the binary combinatorial setting may prove to be a good test ground for understanding this process better.

We also analyzed the prevalence of coding patterns for this case of a DNF with three positive and one negative variable per clause as we did for networks that learn a DNF of clauses with four positive clauses. We ran the same set of tests and benchmarks. We found that the positive witnesses use a 3P1N pattern in which the negative weight in the pattern aligns with the negative variable in the clause. We will call this pattern 3P1Nc, where the $c$ indicates "aligned with the clause," differentiating it from 3P1Nnc, the appearance of a 3P1N pattern where the negative weight is "not aligned with the clause." The left side of Figure 22 shows the 4-AND DNF positive rows coded by 4P for the case $j = 64$ that we already saw in Figure 6 (with all positive variables). The middle of Figure 22 shows the same type of plot for clauses that include a negative variable, and thus include the patterns 3P1Nc and 3P1Nnc. The right hand side shows the corresponding random pattern distribution. As can be seen, the neural network is using 3P1Nc as the clear non-random positive coding pattern for clauses with 3 positives and a negative, with a plot in red that looks very similar to the blue 4P pattern on the right, while the 4P patterns in this case look pretty much random, as we would expect. We do not show the coding in the negative rows, but report that it uses 2P2N for lower $k$'s but eventually saturates as $k$ grows, looking similar to random.

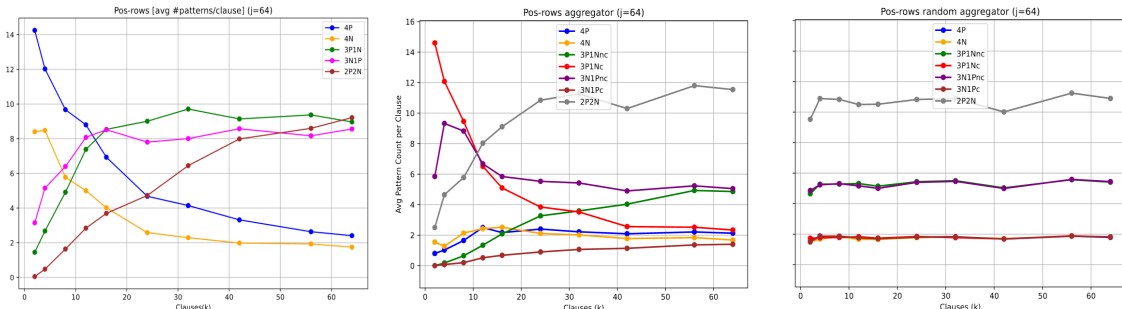

Figure 22: Left panel: a reminder of what clauses with 4 positive variables look like. Middle and right panel: Clauses with 3 positive and one negative variable. The middle panel shows the 3P1Nc coding in the positive rows that scales like the 4P pattern we saw for the 4-AND networks in the graph on the left, and very differently than the random distribution on the right.

## H  FURTHER EXAMPLES OF FEATURE CHANNEL CODING

We next take a look at other examples of problems that the network used feature channel coding to solve: ORs, clauses in CNF form, and a toy vision problem. These help further clarify this techniques versatility and generality.

### H.1  OR AND CONJUNCTIVE NORMAL FORM

We here study the difference in how feature channel coding works between computing an AND and computing an OR. Within the framework of soft Boolean logic, the difference between an AND and an OR is the bias. If we ignore the magnitude of the final result, then $x_1 \wedge x_2 = \text{ReLU}(x_1 + x_2 - 1)$, while $x_1 \vee x_2 = x_1 + x_2$. Thus, we expect gradient descent to find solutions where AND has significant negative bias, whereas OR have minimal bias. To study this, we first look at a very simple Boolean formula, where the output is simply the OR of all the variables in the system. We compare this to the type of formula presented in Figure 1. We studied the same network as in that scenario (hidden dimension equals input dimension equals 16, single neuron at the second layer), and we examined the Layer 1 bias for the neurons at Layer 1 with positive Layer 2 weights associated with them.

|  | Average Layer 1 bias (Mean $\pm$ SD) | Maximum absolute Layer 1 bias (Mean $\pm$ SD) |
|---|---|---|
| AND | -0.61 $\pm$ 0.13 | 1.54 $\pm$ 0.21 |
| OR | 0.0079 $\pm$ 0.023 | 0.0306 $\pm$ 0.085 |

Table 1: Comparison of AND vs OR Layer 1 bias for neurons with positive Layer 2 weight, including standard deviations (SD).

Table 1 summarizes the result of 10 random network trainings, each consisting of 10000 randomly generated inputs (settings of the variables). For AND, the inputs are as described for Figure 1. For OR, each input is a 1 independently with probability 0.043 and 0 otherwise; this provides approximately equal probability of a

positive and negative instance. In this table, we see that the result are as expected: the AND function has a negative bias, which on average is large enough to have an impact, although as already discussed, the value of the bias tends to be a bit smaller than the theory would predict. The OR function, on the other hand, has a very small bias, with a negligible impact on the compuatation.

In an effort to further study the OR function, a seemingly good test bed would be to study formulas in Conjunctive Normal Form (CNF), which is the AND of a number of clauses, where each clause is the OR of a number of literals. The network solves the DNF formulas studied above by computing the AND clauses at the first layer, and then using the single neuron at the second layer to compute an OR of those results, and so our expectation was that the network would solve a CNF formula by computing the OR clauses at the first layer, and then use the second layer to compute the AND of those results. Our expectation was wrong, but we actually found something more interesting.

To study CNF, our neural network setup is the same as the above, with input dimension 16 and hidden dimension 16. We study CNF formulas where every clause is the OR of two (positive) variables, and each variable is used exactly once (the *feature influence* Adler & Shavit (2024) of the network is 1). To train any given formula, we want to construct a roughly even split between True inputs and False inputs; we do this simply by choosing every variable to be True independently with probability 0.75. We studied a number of CNF formulas chosen randomly from the set of such formulas, and saw very consistent results. We here present results for one such randomly chosen formula:

$$(x_3 \vee x_{11}) \wedge (x_9 \vee x_{12}) \wedge (x_2 \vee x_8) \wedge (x_0 \vee x_{10}) \wedge (x_{13} \vee x_{14}) \wedge (x_4 \vee x_7) \wedge (x_1 \vee x_5) \wedge (x_6 \vee x_{15}).$$

In Figure 23, we depict what the network learns for this formula with 40,000 randomly chosen inputs. We see in this figure that the network is not learning the AND as we had expected, and in fact, there is not much that is interesting happening on the small number of positive Layer 2 weight rows at all. On the other hand, the negative rows look much more interesting, and they take up the majority of the neurons. Furthermore, those rows do fit the pattern of feature channel coding, but they also do not look like they are computing the formula as described. On closer inspection, we see that the network has actually learned something clever: it has used a different representation of the same logical function, specifically:

$$\neg \left( \begin{matrix} (\neg x_3 \wedge \neg x_{11}) \ \vee \ (\neg x_9 \wedge \neg x_{12}) \ \vee \ (\neg x_2 \wedge \neg x_8) \ \vee \ (\neg x_0 \wedge \neg x_{10}) \ \vee \\ (\neg x_{13} \wedge \neg x_{14}) \ \vee \ (\neg x_4 \wedge \neg x_7) \ \vee \ (\neg x_1 \wedge \neg x_5) \ \vee \ (\neg x_6 \wedge \neg x_{15}) \end{matrix} \right)$$

This can be seen to be equivalent to our original presentation of the formula by applying De Morgan's Rule at two different levels: once to the individual OR's of the clauses, and then again to the overall AND of the clauses. Of course, the network has no actual knowledge of De Morgan's Rule - it is simply given a subset of the truth table for the formula, and it learns some representation of that truth table. It is interesting to note that based on this example, the network does seem to have a propensity to learn DNF formulas instead of CNF formulas.

In a bit more detail, we see that the network is actually learning this formula as follows. The outer negation is realized through the utilization of the negative Layer 2 weights: if a positive signal comes through the channel codes, then that is converted into a significant negative value by these negative weights. The neuron at the second layer takes the OR of these negative weights: if any one of them evaluates to a (large enough) negative value, then the entire system evaluates to a negative value. If none of them are negative, then the small positive weights on the postive Layer 2 weight rows are enough to make the whole system positive. Each of the clauses is computed through a distinct feature channel code, and relies on the fact that $(\neg x_1 \wedge \neg x_2)$ can be computed in soft lagic as $\text{ReLU}(b - x_1 - x_2)$ where $b = 1$ in the binary case. Due to the negative values of the variables, the codes take on negative values (and so are depicted in red in this figure). We note that the

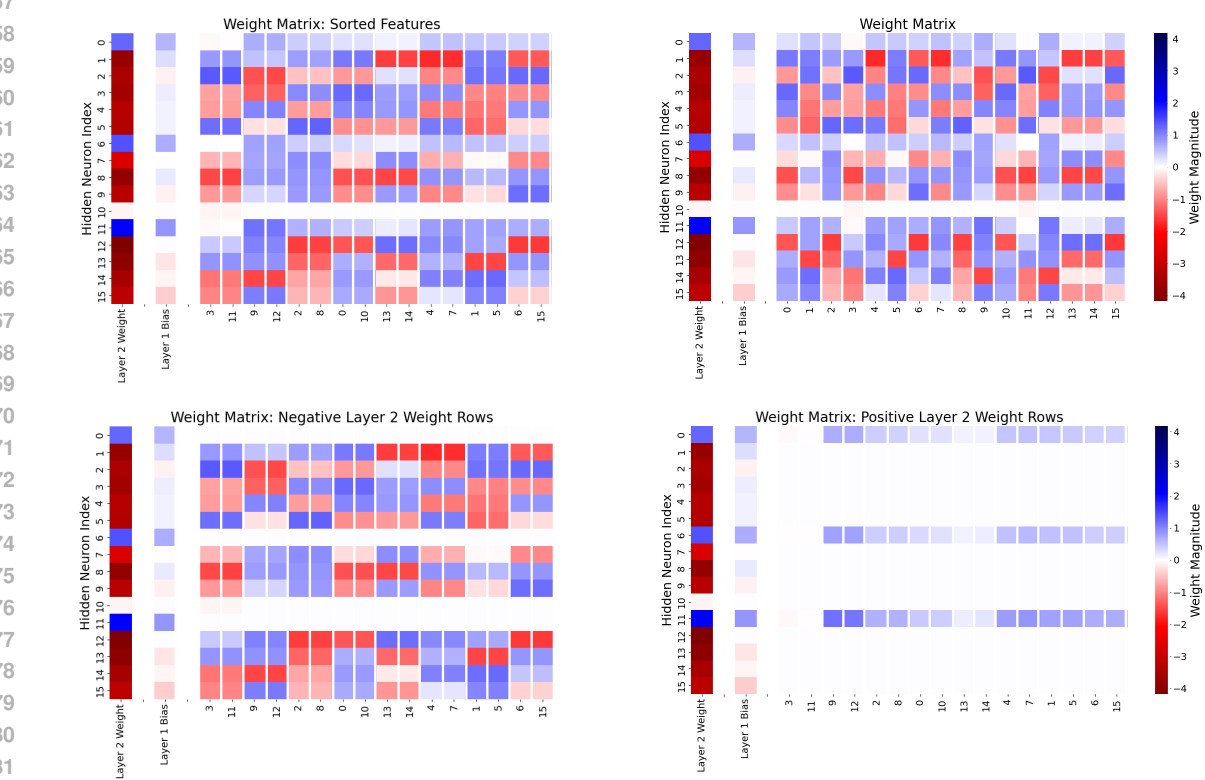

Figure 23: All weights and biases for a neural network trained on the Boolean formula
$(x_3 \vee x_{11}) \wedge (x_9 \vee x_{12}) \wedge (x_2 \vee x_8) \wedge (x_0 \vee x_{10}) \wedge (x_{13} \vee x_{14}) \wedge (x_4 \vee x_7) \wedge (x_1 \vee x_5) \wedge (x_6 \vee x_{15})$

Layer 1 bias is usually positive (corresponding to the +1 in that formula), but it is not usually as large as that formula dictates, which is analogous to what we see with DNF formulas, except with the opposite sign. Here though, some of the rows are negative, which is not analogous to the DNF formulas covered above, where we saw consistently negative bias in the positive weight rows. However, this is compensated for by positive values that appear elsewhere in the row. Due to the way the variables were set for the inputs, there is a high likelihood that every row will have an additional positive contribution, to make up for the slightly smaller bias.

## H.2 CODE ACCURACY IN A ONE DIMENSIONAL VISION PROBLEM

We next turn our attention to a problem that extends beyond Boolean formulas, into a pattern matching problem that could be viewed as a one dimensional vision problem. We here consider the *consecutive four* problem: given an input string of binary values, are there 4 ones in a row? Thus the individual input variables are again binary. We consider this an extremely simple vision problem, in the sense that it deals with pattern matching with locality. With that locality in mind, we we will train and test our network with a restricted set of inputs to this problem: all inputs have exactly six ones in the string, and they must all be close to each

other - within a region of eight consecutive bits of the input string. We want to see if neural networks trained to solve this problem have properties that are consistent with what we see with pure Boolean problems.

We point out that, like many pattern matching problems, this problem can be solved with a Boolean formula: the OR of all four ways ANDs of four consecutive bits. And in fact, even though the training process is not given this representation (only the ordered sets of inputs and their labels), using the Combinatorial Interpretability approach, we see that the networks we train to solve this problem do in fact use exactly that formula.

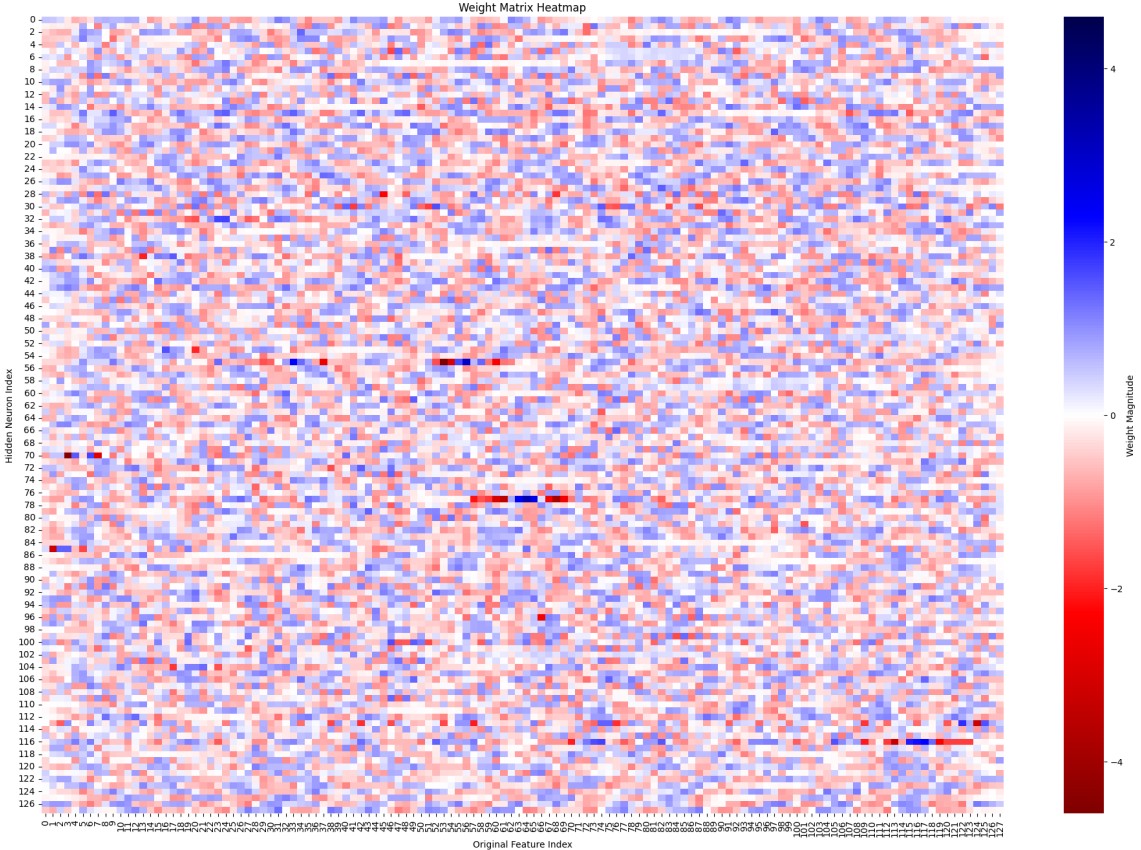

Figure 24: Layer 1 heat map for the connect four problem. The x-axis corresponds to the pixels of the one dimensional image we the network receives as input.

Our setup is as follows. We use a network with 128 input variables, 128 hidden neurons in the single hidden layer, and a single output neuron. We generate each training and test data input as follows:

- Flip a fair coin for each input to determine if it will be positive or negative.
- In either case, choose a sequence of eight consecutive variables uniformly at random.
- For a positive input, out of the set of eight chosen variables, pick a consecutive set of four variables uniformly at random. Also choose two other variables from the eight uniformly at random. These six variables are set to 1, and all others are set to 0.

- For a negative input, randomly pick a consecutive set of four variables out of the eight, but only set three of them to 1. Then pick another three of the eight variables uniformly at random, and set those variables to 1 as well. Then test if the result has four consecutive variables set to 1. If it does, repeat until a negative input is generated, or 20 attempts have failed, in which case use the string of all zeros.

We generated 10,000 training inputs, and trained for 20 epochs, after which the network had 0 loss. Accuracy on the test set was 99%. The resulting weight matrix, shown in Figure 24, at first glance, might not seem to exhibit channel coding.

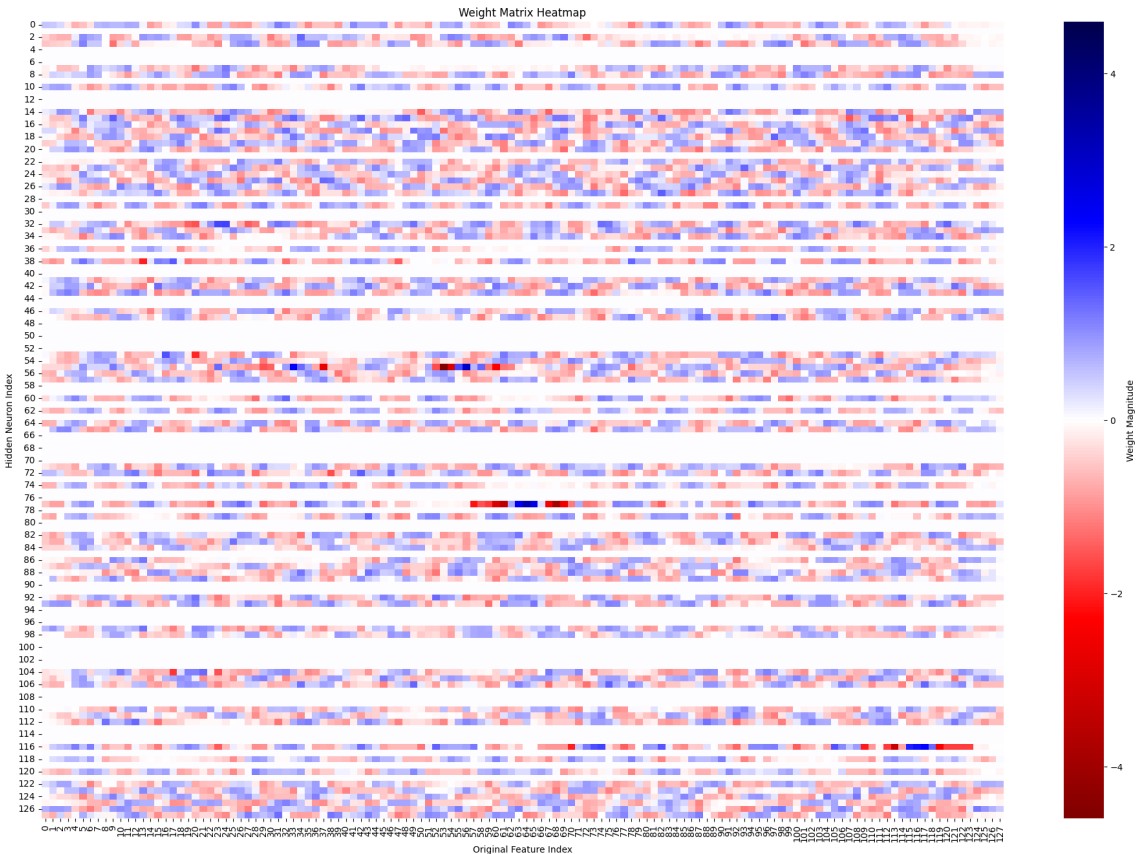

Figure 25: Layer 1 for connect four: positive witnesses.

However, things become quite a bit clearer when we separate the rows into positive witnesses and negative witnesses, where (as before) a row is considered a positive witness if its Layer 2 weight is positive, and a negative witness if its Layer 2 weight is negative. The positive witness only heat map is depicted in Figure 25. This is a subset of the rows shown in Figure 24; no other changes have been made.

We can see visually that the positive witness rows do in fact demonstrate considerable feature channel coding. And in fact, we see that most of the rows of the (positive only) matrix consist of alternating sequences of four consecutive positive values, followed by a number of negative values. There are some sequences of ones that are longer than four, but most of them have length exactly four. This image alone provides convincing

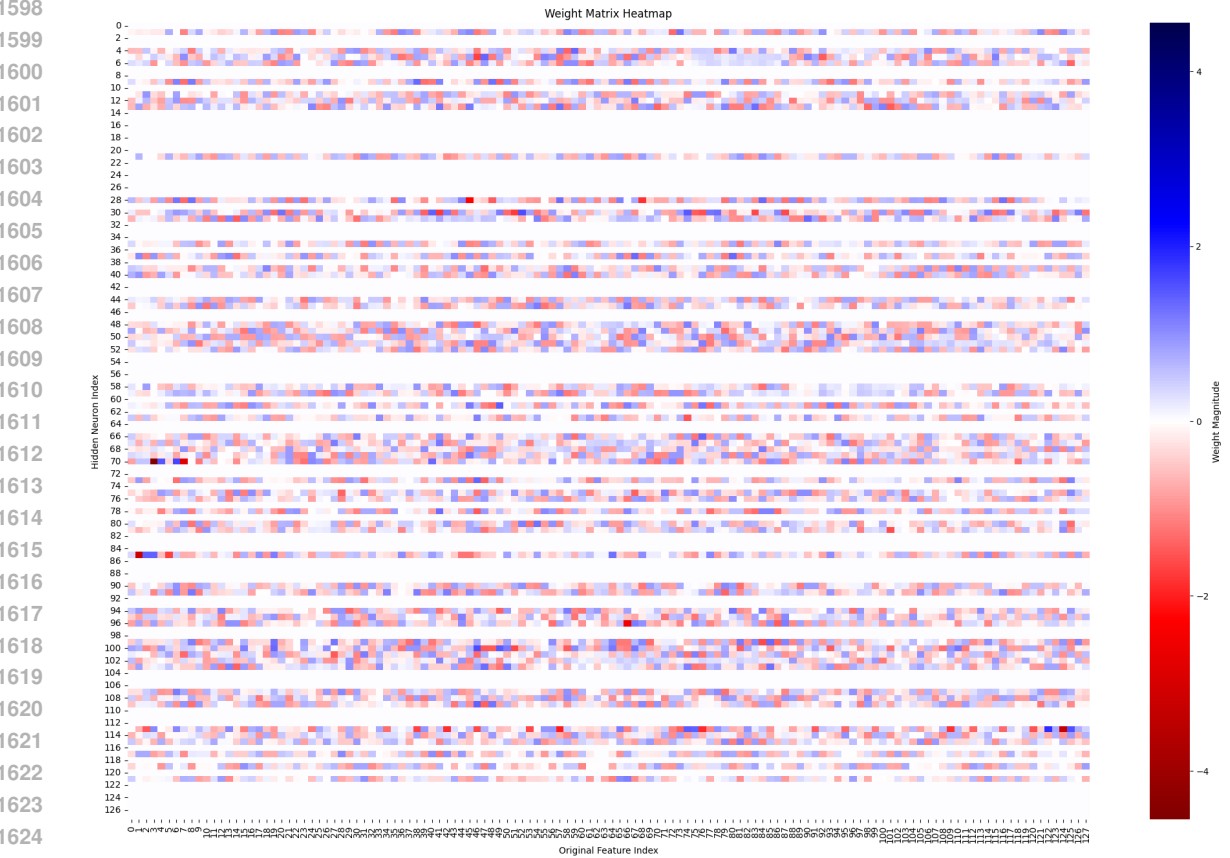

Figure 26: Layer 1 for connect four: negative witnesses.

evidence that this network uses feature channel coding, but we will demonstrate this with metrics as well. In this context, we consider a row to be participating in the code for a sequence of four consecutive ones starting at position $i$, if that row (a) corresponds to a positive layer 2 weight, and (b) it has four consecutive positive values starting at position $i$. The resulting statistics are summarized in Table 2.

These statistics clearly indicate that coding is being used here: an average of 12.77 rows code each possible positive example, and there is only an average overlap of 2.35 between rows. Furthermore, all coding channels clearly express a compution closely resembling an AND of the four bits that are being coded for: every single coding row has a negative bias (even though the biases are initialized uniformly and independently at random between -1 and 1).

The negative witnesses are depicted in Figure 26. We see here that the negative witnesses consist of alternating sequences of one positive value and one negative value, essentially computing an XOR.

We next show that we can extract and perform classification using the codes that are depicted in Figure 25, and measured in Table 2. We can view this process as a combinatorial replacement (for this specific network) of a sparse autoencoder: these codes correspond to how each feature is represented by the neural network by its positive coding rows. A feature here is any set of four consecutive ones in the input string. Extracting

| Statistic | Average | Std Dev |
|---|---|---|
| **Codingness Statistics** | | |
| Total Number of Unique Coding Rows | 70 | - |
| Fraction of Contributions from Coding Rows | 63.93% | 10.29% |
| Number of Coding Rows | 12.77 | 2.73 |
| Weighted Sum of Coding Rows | 9.0139 | 2.1991 |
| **Overlap Statistics** | | |
| Number of Overlapping Coding Rows | 2.35 | 1.64 |
| Weighted Overlapping Coding Rows | 18.24% | 12.71% |
| **ANDness Statistics** | | |
| Percentage of Negative L1 Bias Coding Rows | 100.00% | 0.00% |

Table 2: Summary of Codingness, Overlap, and ANDness

the codes is straightforward: we simply sweep across the different starting positions, and for each starting position extract the codes for that position per the definition of coding rows.

To use these extracted codes for classification, we construct a decoding algorithm that maps the Layer 1 post-activation values in positive rows of any input to a set of features that are active. In other words, the algorithm attempts to make a decision as to where there are four consecutive ones in the input string, not based on the entire neural network, but instead only on the Layer 1 post-activation values, and of those values, only those that are associated with positive witnesses. This serves three purposes: (1) it further demonstrates how to combinatorially interpret the trained model, (2) it demonstrates how well we can do with only that subset of information (and specifically without using any of the negative witnesses), and (3) it also demonstrates that the network is really learning the actual underlying features, as opposed to the summary (binary) classification problem it was asked to solve.

The algorithm proceeds by computing the Layer 1 post-activations of the network, and then comparing these values to the codes. Any feature is considered to be present if its code is entirely present. A code is entirely present if all of its rows have a post-activation value that is at least the sum of its four positive elements, minus its bias, minus 1.9 times the maximum value within its sliding window of eight consecutive bits. We test this algorithm on a set of inputs drawn from the same distribution as both the training set and the test set of inputs, and find that this very simple decoding algorithm does a surprisingly good job of not just differentiating 0 inputs from 1 inputs, but also determining which features are actually present. The results from a single training run (and accompanying extraction of codes), tested on a set of 40 random test inputs is provided in Table 3. The actual matches are found simply by scanning the input for four consecutive ones; the code matches are from our extracted codes.

We test this more thoroughly by running 50 different training runs, and for each training run testing the accuracy of the algorithm's predictions on a test set of 2000 inputs, drawn from the same distribution as the training set. We measure success by two criteria: how well did the algorithm predict whether or not there is a matching pattern of four consecutive ones, and how well did the algorithm predict the exact set of matches (which is the empty set if there is no match). Our results are summarized and compared to the original trained network in Table 4.

We note that while the results of our algorithm are noticably worse than the original network, it is still over 92% accurate in terms of the decision problem, and over 82% accurate in terms of picking out the exact set of matches (which the original network does not do). These results are important for a number of reasons:

| Test | Label | Actual Matches | Code Matches |
|------|-------|----------------|--------------|
| 0 | 1 | [35] | [35] |
| 1 | 0 | — | — |
| 2 | 1 | [88, 89] | [88, 89] |
| 3 | 1 | [116, 117] | [116, 117] |
| 4 | 1 | [11, 12] | [12] |
| 5 | 0 | — | — |
| 6 | 0 | — | [116] |
| 7 | 1 | [18, 19, 20] | [18, 19, 20] |
| 8 | 0 | — | — |
| 9 | 1 | [40] | — |
| 10 | 1 | [15, 16] | [16] |
| 11 | 1 | [14, 15, 16] | [14, 15, 16] |
| 12 | 0 | — | — |
| 13 | 0 | — | — |
| 14 | 1 | [47, 48, 49] | [47, 48, 49] |
| 15 | 0 | — | — |
| 16 | 0 | — | — |
| 17 | 0 | — | — |
| 18 | 1 | [97] | — |
| 19 | 1 | [62, 63] | [62, 63] |

| Test | Label | Actual Matches | Code Matches |
|------|-------|----------------|--------------|
| 20 | 1 | [4] | [4] |
| 21 | 1 | [106, 107] | [106, 107] |
| 22 | 0 | — | — |
| 23 | 1 | [29, 30, 31] | [29, 30] |
| 24 | 0 | — | — |
| 25 | 0 | — | — |
| 26 | 0 | — | — |
| 27 | 1 | [8, 9, 10] | [8, 9, 10] |
| 28 | 0 | — | — |
| 29 | 1 | [24, 25, 26] | [24, 25, 26] |
| 30 | 1 | [69] | [69] |
| 31 | 0 | — | — |
| 32 | 0 | — | — |
| 33 | 0 | — | — |
| 34 | 1 | [80, 81, 82] | [80, 81, 82] |
| 35 | 0 | — | — |
| 36 | 0 | — | — |
| 37 | 1 | [62] | [61] |
| 38 | 0 | — | — |
| 39 | 0 | — | — |

Table 3: Sample output from testing: actual matches versus code matches.

| | FPR NN | FNR NN | FPR CODES | FNR CODES | FULLY CORRECT |
|--------|--------|--------|-----------|-----------|---------------|
| Average | 0.71% | 1.65% | 6.82% | 7.66% | 82.63% |
| Std Dev | 0.33% | 0.37% | 1.82% | 0.97% | 1.80% |

Table 4: Performance of Coding Interpretability Compared to Original Neural Network.

- They further demonstrate how central the codes are to the computation.

- Even though the algorithm described above only uses the positive witnesses, they are able to achieve over 92% accuracy on the decision problem of whether there is a match or not.

- Even though the neural network was trained just to answer the decision problem, looking at the network through the lens of the resulting codes yields over 82% accuracy on precisely classifying where all the consecutive four sequences are. As we see from Table 3, often when it is wrong, it is very close to providing the correct answer (i.e., a sequence of 5 consecutive ones should return two matches, but the coding-based algorithm only returns the first one.)

We next turn to another phenomena we observed in studying this problem. This is not directly relevant to our central hypothesis of networks using feature channel coding, but it does demonstrate what can be discovered by using our combinatorial interpretability approach. To see this phenomena, we looked at all columns of the layer one weight matrix, and we computed all pairwise correlations between them. This is depicted in Figure 27. The strong self correlation on the diagonal is of course expected, and we see an expected pattern in the values close to that diagonal. We did not, however, expect to see such a strong and regular pattern in the upper and lower diagonal regions of this matrix, even for columns that represent inputs that are very far removed from each other.

Our best guess for the cause of this pattern is that there is a strong preference for a new set of four consecutive ones to appear on a different neuron after the current one has finished any existing pattern. This would cause regions of alternating sequences of "more likely to code" and "less likely to code", each of length four, which is consistent with the pattern. This preference presumably is fairly strong for it to not perceptibly fade throughout the entire matrix.

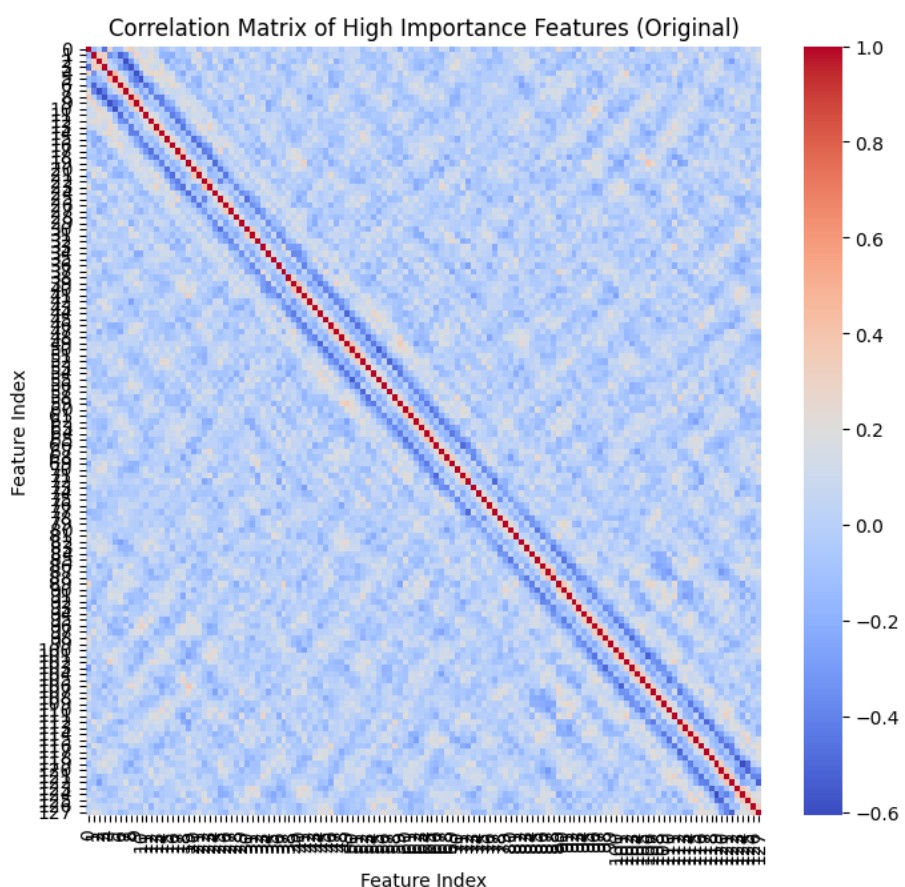

Figure 27: Correlation Matrix for columns of layer 1 weight matrix.

### H.3 POLYSEMANTICITY AND SUPERPOSITION

Let us touch a bit more on the notions of polysemanticity and superposition Adler & Shavit (2024); Elhage et al. (2022); Hänni et al. (2024); Henighan et al. (2023). When one looks at the set of neurons of a feature channel code as in our example above, we see that there is overlap. The same neuron is polysemantic and fires in response to multiple features. The accepted idea for how superposition is implemented is that neurons represent features using polysemanticity. To quote Elhage et al. (2022): "in the superposition hypothesis, features are represented as almost-orthogonal directions in the vector space of neuron outputs. Since the features are only almost-orthogonal, one feature activating looks like other features slightly activating." This is true, and yet our theory of feature channel coding suggests that superposition can also be explained by the combinatorial property of using feature channel codes that overlap. The noise of one feature slightly lighting up others is explained not just by alignment in vector space, but also by the use of multiple rows in the feature channel coding so the relative importance of a given neuron is lowered Adler & Shavit (2024). Moreover, we don't need to decipher directionality in activation space or geometric representations in order to map the features captured by a collection of neurons: the feature channel coding in the weight matrices allows us to do it combinatorially without any activation information. Furthermore, as we saw in Section C, using the

combinatorial interpretation also allows us to analyze the code pattern distributions of networks to explain scaling laws in ways that seem impossible to capture by looking at features through their directionality in activation space.

Getting back to our example network described in Section 2, if overlapping codes introduce noise, given that our network was trained on 8 clauses using 16 neurons in the first layer, the training could have found a coding that had one code row per clause: one monosemantic neuron per feature. In this way, there would be no overlaps among codes, and thus no cross-feature noise. However, as in the matrices resulting from our training, we see that the training settled on multiple overlapping codes and the neurons are polysemantic. In other words, polysemanticity appears even when there are fewer features than neurons.

We believe that there is a combinatorial explanation for this. Gradient descent searches the state space for local minima. There are many more encodings of the desired computation using overlapping codes than using non-overlapping monosemantic ones, and thus likely many more local minima involving feature channel coding using multiple overlapping polysemantic neurons. Thus, gradient descent is more likely to find such an encoding than the monosemantic one, and then not be able to move from that local minimum. This is somewhat reminiscent to the findings of Lecomte et al. (2024). As witnessed in our example, coding (and with it polysemanticity) does appear in cases where one does not need to compute in superposition. It would be interesting to obtain more evidence or a proof of this hypothesis, which could be done using an approach similar to the one we take in the work that produced Figure 5.

# I    FURTHER RESEARCH

We believe the combinatorial approach and its first application through feature channel coding, can have implications even before we have the ability to apply it to large real world networks. In particular, in the same way we used it to understand scaling laws, one can attempt to understand other important neural computation aspects such as sparsity, quantization, and the linear representation hypothesis, to name a few. Understanding to what extent computation in real production neural networks contains Boolean features and tehir codes is also of great interest. Finally, it has not escaped our notice that the specific coding we have postulated immediately suggests a possible computation mechanism for neural tissue.

