# OpenReview forum: "Toy Models of Combinatorial Interpretability"
_ICLR.cc/2026/Workshop/Sci4DL — Sci4DL 2026_

### Official Review · Reviewer_kiEX · 2026-02-26

**Fit:** 2
**Significance:** 2
**Confidence:** 2

**Summary:**

This work looks at interpreting networks trained to fit boolean functions. The sign of the weights are interpreted to infer the underlying boolean function that the network learned.

**Strengths:**

The paper is pretty well-written and reads well. I like that they have an appendix with more general results. I'm afraid I didn't have the ability to read through it in much detail.

Probably one of the more exciting claims are about the scaling laws, where error decreases in hidden layer size.

**Suggestions:**

The authors don't spend much time discussing the limitations of just looking at the signs of the weights as opposed to their magnitude. It would seem that this approach only applies for boolean functions, not more general functions.

The results in Fig 1 seem to require advance knowledge of the structure of the boolean function learned to group the variables as you did. Therefore, it's not clear to me the significance of the results for understanding learning of functions where the targets are data-derived.

Overall, the scaling law conclusions seem based on observed correlations more than by any theory or strong evidence.

There isn't any reference to the classic work by McCulloch and Pitts on expressibility of boolean functions with neural networks. Their paper is where the simple AND function and other logical functions were first explained. I guess I get the feeling, while reading this work, that it's something that I've seen before.

Maybe it's the format, but overall I'm not sure exactly what I should take away from these results. A short discussion at the end of the main paper would help clarify and simplify the overall message.

* Many references should be parenthetical but aren't
* line 080: why are there $i$ subscripts here?
* line 108: "largely inactive" it would seem that "unused" is a better word here since you're talking about weight not activity
* clearly define what you mean by 4P, 3N1P codes
* In most of the supp figures, text was very small and unreadable even zoomed in on my screen. I suggest changing the font sizes and image formats to avoid pixelation.
* Fig 4 appears to be a screenshot and has a "Plot area" box
* Is Fig 7 is the same as Fig 3?

---

### Official Review · Reviewer_guU8 · 2026-02-27

**Fit:** 3
**Significance:** 2
**Confidence:** 2

**Summary:**

The main paper studies the internal structure of a single-layer, single-output MLP using a Boolean function consisting of a sequence of AND and OR operations. It views features as encoded in subsets of neurons, and studies these encodings along with the category of weights and biases (in this case, positive or negative) rather than their precise size. It shows how the Boolean function can be found in the pattern of positive weights and negative biases, encoding the expected ‘AND’ features, when one knows what the function is to look for.

**Strengths:**

Results in the main paper are light and somewhat anecdotal but the appendix hides additional detail (more detailed descriptions, background, multi-layer networks.)

Some interesting ideas are presented. There is a clear fit with the workshop, and this work should elicit some good discussion.

**Suggestions:**

Main comments:
- It would be very useful if this work could be contrasted more directly with the closest variant of circuit analysis.
- In general, the paper tends to make sweeping statements that are not fully true: “It also allows us for the first time to exactly quantify and explain the relationship between a network’s parameter size and its computational capacity”; “Specifically, our theory explains phenomena like polysemanticity and superposition as arising from feature-associated codes”; etc. Please review these.
- The current appendix can benefit from a reorganisation. Ideas can be presented more logically and sequentially. 30 pages is a bit extensive in support of the main paper.

Some minor detail:
- Caption of Fig 1 can include more information. (Explain sub-figs and ordering.)
- Fig 2: link between legend and graph is mostly missing.
- “The network was trained to learn various hidden Boolean functions.”  This seems to imply that various functions are used to generate training data for a single network, but it later becomes clear that a single function is used per network trained. (Is that correct?)
- “We give a formal definition of feature channel coding in the appendix” -> Consider pointing to E3 and clarifying the description there. The description is currently unnecessarily dense. Actually, this description would be useful earlier on, in the main paper, if space allows.
Introduce abbreviations.
- “patetrn”
- What does “totally non-random” mean?

---

### Meta-Review · Area_Chair_h6QV · 2026-03-01

**Recommendation:** Accept

**Metareview:**

Recommending accept, a decent fit for the workshop and interesting work.

---

### Decision · Program_Chairs · 2026-03-02

Accept